# A conformable fractional finite difference method for modified mathematical modeling of SAR-CoV-2 (COVID-19) disease

**Syeda Alishwa Zanib**[1], **Tamour Zubair**[2]*, **Sehrish Ramzan**[3], **Muhammad Bilal Riaz**[4], **Muhammad Imran Asjad**[5], **Taseer Muhammad**[6]

**1** Department of Mathematics, Riphah International University, Faisalabad, Pakistan, **2** School of Electrical and Electronic Engineering, The University of Adelaide, Adelaide, Australia, **3** Department of Mathematics, Government College University Faisalabad, Faisalabad, Pakistan, **4** Department of Computer Science and Mathematics, Lebanese American University, Byblos, Lebanon, **5** Department of Mathematics, University of Management and Technology, Lahore, Pakistan, **6** Department of Mathematics, College of Science, King Khalid University Saudi Arabia, Abha, Saudi Arabia

* tamour.zubair@adelaide.edu.au

**Data Availability Statement:** Data sharing is not applicable to this article as no datasets were generated during the current study.

## Abstract

In this research, the ongoing COVID-19 disease by considering the vaccination strategies into mathematical models is discussed. A modified and comprehensive mathematical model that captures the complex relationships between various population compartments, including susceptible ($S_\alpha$), exposed ($E_\alpha$), infected ($U_\alpha$), quarantined ($Q_\alpha$), vaccinated ($V_\alpha$), and recovered ($R_\alpha$) individuals. Using conformable derivatives, a system of equations that precisely captures the complex interconnections inside the COVID-19 transmission. The basic reproduction number ($R_0$), which is an essential indicator of disease transmission, is the subject of investigation calculating using the next-generation matrix approach. We also compute the $R_0$ sensitivity indices, which offer important information about the relative influence of various factors on the overall dynamics. Local stability and global stability of $R_0$ have been proved at a disease-free equilibrium point. By designing the finite difference approach of the conformable fractional derivative using the Taylor series. The present methodology provides us highly accurate convergence of the obtained solution. Present research fills research addresses the understanding gap between conceptual frameworks and real-world implementations, demonstrating the vaccination therapy's significant possibilities in the struggle against the COVID-19 pandemic.

## 1 Introduction

In the modern world, epidemics like Ebola, *HIV*, *HBV*, $H_1N_1$, and malaria are receiving more attention over time, and it is difficult to stop diseases from spreading among the populace. On the other hand, the globe continues to battle already-existing infectious diseases, while on the other sight, shifting global circumstances promote the birth of various viral kinds. The coronavirus shown in Fig 1, which first surfaced in early 2020 and is still not completely under

**Funding:** The authors received no specific funding for this work.

**Competing interests:** NO authors have competing interests.

control, is the newest and most potent of these viruses in recent years. While the first instances were found in Wuhan, China, on December 31, 2019, [1–3], the disease's biological cause has not yet been fully identified. Lung disease has a high mortality rate and may be found all over the world because the World Health Organization (WHO) has declared it a pandemic. If left untreated, it can also lead to the spread of viruses that cause diseases like severe acute respiratory syndrome. The three coronavirus subgroups are alpha, beta, and gamma. SARS-CoV is a member of a fourth new class of viruses known as delta coronaviruses. In the middle of the 1960s, human coronaviruses were first discovered [3].

In its broadest definition, mathematical modelling is an attempt to use mathematics to explain a phenomenon, an event, and the relationships between them without using mathematics, or it is the process of finding mathematical techniques within these phenomena and occurrences [4]. Through the process of modelling, mathematics is an organized way of thinking that produces answers for occurrences and issues that happen in the actual world. We see that the fundamental principles of mathematical notions have their roots in real phenomena and connections to them when mathematics is applied to the world. Modelling approaches and solution techniques for these problems need to be created because many problems, particularly today, are complicated, non-linear, have memory effects, or have stochastic structures. Although mathematical models cannot offer treatment for a specific infectious disease, they can be used to illustrate and examine potential outcomes of the current dynamics [5]. In a short period, numerous investigations on the mathematical model of the COVID-19 pandemic have been published in the literature. In 2020, Zu et al. [6] created and tested COVID-19 contagion models on the Chinese mainland as well as the efficacy of various control measures. Then, using the sensitivity analysis method, they were able to forecast the efficacy of various intervention options while effectively estimating the epidemic trend and COVID-19

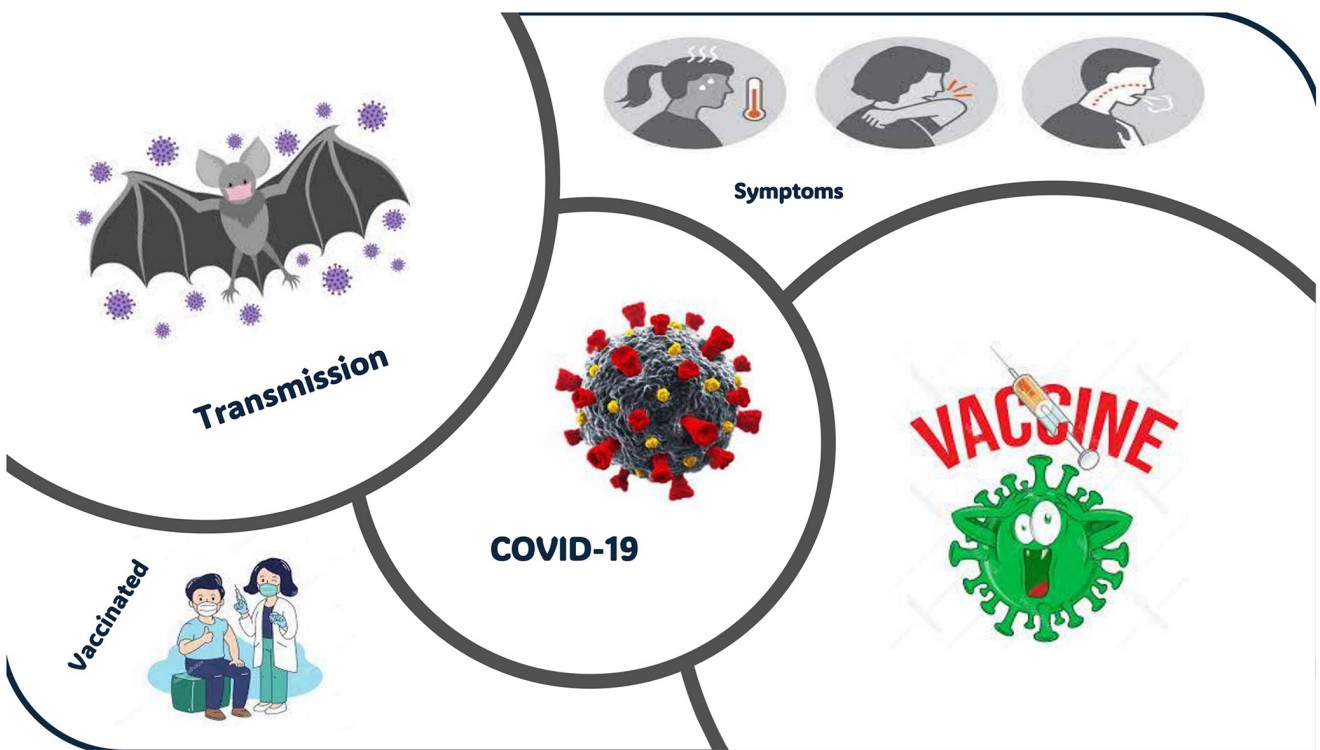

**Fig 1. COVID-19 transmission, symptoms and vaccination.**

transmission risk. The COVID-19 transmission in China's final phase of the pandemic was discussed in 2020 by Tang et al. [7] analysis of the efficacy of isolation and quarantine. According to their results, effectively controlling the COVID-19 epidemic has needed further improvements to isolation and quarantine procedures as well as higher detection rates in mainland China. Uncertainty studies that highlight the continuous unpredictability of the epidemic show that these measures have been crucial in the nation's response. It is crucial to continue working on these techniques to resolve the remaining uncertainties. In 2020, Ahmed et al. [8] used numerical methods and logistic models to analyze a mathematical model of COVID-19, they introduced and analyzed a few COVID-19 models that contain crucial queries regarding international health care and provide crucial suggestions. They suggested using Euler's method, the second order (RK2), and the fourth order (RK4) Runge-Kutta methods to solve the given equations. In 2020, using numerical simulations, Okuonghae and Omame [9] evaluated the effects of control measures on COVID-19 dynamics, focusing on social distance, face mask use, and testing. They also produced important predictions for the total number of reported cases and the various intensities of control measures used. According to numerical simulations of the model, the disease would eventually vanish from the population if at at least 55 per cent of the population agreed with the social distance limit and roughly 55 per cent of the population utilized face masks efficiently in society. In 2021, Srivastav et al. [10] investigated the COVID-19 pandemic's behavior in India, to evaluate the effects of the face mask, hospitalization of symptomatic patients, and quarantining of asymptomatic people. They discovered that hospitalization of symptomatic patients, isolation of asymptomatic patients, and regular use of face masks in public places were all useful strategies to decrease the impact of COVID-19 in India. In 2021 Yavuz et al. [11] developed a mathematical model to reveal the effects of vaccine treatment, which has been performed recently, on COVID-19 in this study. In their proposed model, as well as the vaccinated individuals, a five-dimensional compartment system including the susceptible, infected, exposed and recovered population was constructed. According to the research by Pearson et al. [12] in 2021 if the COVID-19 vaccine was reasonable and highly effective, that could be cost-effective even in low and middle-income populations. In 2021, Alzahrani et al. [13] developed a model by taking into consideration the environmental contributions of the latent, infected and asymptomatic infected population. The model under consideration is taken in the form of a fractional order ABC derivative. In 2023, Suganya & Parthiban [14] reviewed a mathematical model look at the quantitative analysis and dynamical behaviors of a novel coronavirus, with a focus on the Caputo fractional derivative. In 2023, Jose et al. [15] studied a deterministic mathematical model for Dengue Fever (DF) and Zika virus (ZIKV) co-infection transmission dynamics was formulated and analyzed. In 2023, Jose et al. [16] developed a mathematical model depicting the transmission dynamics of Chickenpox by incorporating a new parameter denoting the rate of precautionary measures. In 2023, Ouncharoen et al. [17] explored a nonlinear SEIR model for COVID-19 transmission dynamics, investigating its stability, reproduction number, and simulations using classical and fractional order methods. Graphical representations accompany the study's findings. In 2024, Abdulwasaa et al. [18] addressed the intricate link between poverty and corruption by developing a mathematical model. Through linear analysis and Eviews software, indicators are examined, leading to predictions of poverty rates. The model, framed with Caputo fractional derivatives, undergoes nonlinear analysis and numerical simulations, with comparisons to real data for validation. Various mathematical models are developed to observe biological diseases using Ordinary Differential Equations (ODEs) as a framework [19–22] Jumarie defined a few basic derivative formulae for fractional calculus in [23], by proposing Modified R-L fractional derivative [24]. Afterwards, in [25–27], a few conflicts regarding Jumarie formulae were raised. So, to resolve those problems, a new definition of fractional

derivatives was defined by Khalil et al., in [28]. We have used the above definition, to study the model of alcohol consumption in Spain, which is very helpful in better understanding the model. In 2013, Mickens et al. [29] examined how conservation laws restrict finite difference discretization for coupled population systems using Mickens' nonstandard finite difference (NSFD) methodology. They identify various conservation law types and illustrate NSFD discretization through popular population models, highlighting their importance in numerical integration challenges. In 2023, Obiajulu et al. [30] analyzed a novel fractional-order mathematical model Using efficient finite difference methods, controlling the co-circulation of dengue and COVID-19, ensuring solution uniqueness via Banach's fixed-point theorem and stability analysis around the infection-free equilibrium. Numerical solutions with the NSFD approach converge to disease-present or -free equilibrium, regardless of initial conditions or fractional orders. During the literature study, adding further compartments to the model, such as those that represent vaccinated and quarantined individuals, can produce more accurate findings. This is because both vaccination and quarantine when considered separately, have the potential to affect the disease's spread and management significantly. The model may better reflect the dynamics of the real-world scenario and give more precise insights into the efficacy of these measures. We will also design the finite difference approach of the conformable fractional derivative using the Taylor series. This numerical method will give us the high convergence solution of the system of equations which is the main objective of this study. **Section 2:** This section will cover the modified COVID-19 transmission model with quarantine and vaccination class. **Section 3:** The reproduction number will be found by using the Next-generation Method of the modified model and also checking its sensitivity analysis. Local and global stability at disease-free equilibrium is also discussed in this chapter. We present the existence of a solution and its uniqueness. **Section 4:** In this section, we approximate the finite difference method of conformable derivative. After discretization, the COVID-19 model's results and discussions will be discussed. **Section 5:** This section will cover the conclusion.

## 2 Model formulation

The provided set of equations represents a mathematical model for the dynamics of a population concerning the spread of a disease, possibly COVID-19. This model is compartmental, specifically an $S_\alpha E_\alpha Q_\alpha U_\alpha V_\alpha R_\alpha$ model, where individuals are categorized into different compartments based on their health shown in Fig 2.

1. Susceptible ($S_\alpha$) are individuals in this compartment who are not infected.

2. Exposed ($E_\alpha$) are individuals in this compartment who have a disease-causing pathogen in their bodies but are not showing any overt clinical symptoms.

3. Infected ($U_\alpha$) are individuals who become infectious and can spread the disease to others.

4. Quarantine ($Q_\alpha$) those who are infected but do not have any viral symptoms.

5. Vaccinated ($V_\alpha$) those who are vaccinated.

6. Recovered ($R_\alpha$) whose are recovered.

$$\frac{dS_\alpha}{d\rho} = \beta - (a_1 E_\alpha + a_2 + d)S_\alpha, \tag{1}$$

The Eq (1) describes the rate of change of susceptible individuals. It includes factors such as

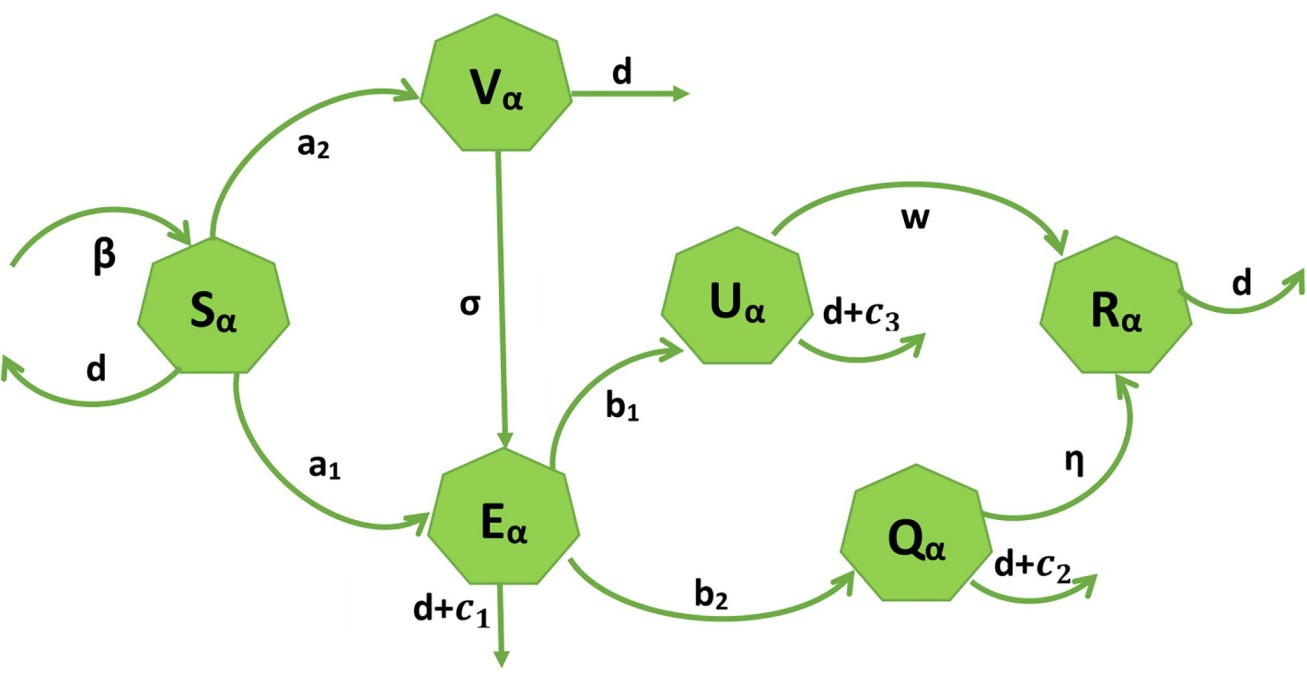

**Fig 2. COVID-19 model.**

the natural birth rate ($\beta$), the transmission from exposed to susceptible individuals ($a_1 E_\alpha$), those who have not been exposed to the disease ($a_2$), and the natural death rate ($d$).

$$\frac{dE_\alpha}{d\rho} = a_1\,E_\alpha S_\alpha + \sigma\,V_\alpha E_\alpha - (b_1\,U_\alpha + b_2\,Q_\alpha + d + c_1)E_\alpha, \qquad (2)$$

The Eq (2) represents the rate of change of exposed individuals. It considers the transmission from susceptible to exposed individuals ($a_1 E_\alpha S_\alpha$), the impact of vaccination ($\sigma V_\alpha E_\alpha$), and factors such as the progression to active infection, quarantine, natural death, and death due to the disease ($b_1 U_\alpha + b_2 Q_\alpha + d + c_1$).

$$\frac{dQ_\alpha}{d\rho} = b_2 E_\alpha Q_\alpha - (\eta + d + c_2)Q_\alpha, \qquad (3)$$

The Eq (3) describes the rate of change of individuals in the quarantine compartment. It includes terms representing the movement of exposed individuals to the quarantine compartment ($b_2 E_\alpha Q_\alpha$) and the factors influencing the exit from the quarantine compartment, such as the recovery rate ($\eta$), natural death rate ($d$), and death due to the disease in the quarantine compartment ($c_2$).

$$\frac{dU_\alpha}{d\rho} = b_1 E_\alpha U_\alpha - (w + d + c_3)U_\alpha, \qquad (4)$$

The Eq (4) represents the rate of change of individuals in the infectious compartment. It considers the transmission from exposed to infectious individuals ($b_1 E_\alpha U_\alpha$) and the factors influencing the transition out of the infectious compartment, including the recovery rate ($w$),

natural death rate ($d$), and death due to the disease in the infectious compartment ($c_3$).

$$\frac{dV_\alpha}{d\rho} = a_2\, S_\alpha - (\sigma\, E_\alpha + d)\, V_\alpha, \tag{5}$$

The Eq (5) describes the rate of change of vaccinated individuals. It includes terms representing the movement of individuals who have not been exposed to the disease to the vaccinated compartment ($a_2 S_\alpha$) and factors influencing the exit from the vaccinated compartment, such as the impact of exposure to the disease ($\sigma E_\alpha$) and natural death rate ($d$).

$$\frac{dR_\alpha}{d\rho} = \eta\, Q_\alpha + w U_\alpha - d R_\alpha. \tag{6}$$

The Eq (6) represents the rate of change of individuals in the recovered compartment. It includes terms representing the movement of individuals from the quarantine compartment to the recovered compartment ($\eta Q_\alpha$), individuals from the infectious compartment to the recovered compartment ($w U_\alpha$), and factors influencing the exit from the recovered compartment, such as natural death rate ($d$) discussed in Table 1.

We will utilize the Khalil conformable derivative, as defined in [28], to explore the memory effects within the model.

$$\frac{d^\phi C}{d\rho^\phi} = \lim_{\xi \to \infty} \frac{C(\rho + \xi \rho^{1-\phi} - C(\rho))}{\xi}, \quad \forall \rho > 0. \tag{7}$$

If C is differentiable the,

$$\frac{d^\phi C}{d\rho^\phi} = \rho^{1-\phi} \frac{dC}{d\phi}. \tag{8}$$

**Table 1. Description of physical parameters.**

| Parameter | Description |
|---|---|
| $\beta$ | Natural birth rate. |
| $a_1$ | Propagation from susceptible individuals to infected individuals. |
| $a_2$ | Transition rate from susceptible to exposed individuals. |
| $\sigma$ | Transition rate from exposed to infected individuals. |
| $b_1$ | Transition rate from infected to symptomatic individuals. |
| $b_2$ | Transition rate from symptomatic to quarantined individuals. |
| $c_1$ | Death rate due to COVID-19 in exposed individuals. |
| $c_2$ | Death rate due to COVID-19 in quarantined individuals. |
| $c_3$ | Death rate due to COVID-19 in infected individuals. |
| $\eta$ | Transition rate from quarantined to recovered individuals. |
| $w$ | Recovery rate of infected individuals. |
| $d$ | Natural death rate. |

To establish the following conformable model of COVID-19 as described in Eqs (1)–(6):

$$\rho^{1-\phi}S'_\alpha = \beta - (a_1 E_\alpha + a_2 + d)S_\alpha,$$

$$\rho^{1-\phi}E'_\alpha = a_1 E_\alpha S_\alpha + \sigma V_\alpha E_\alpha - (b_1 U_\alpha + b_2 Q_\alpha + d + +c_1)E_\alpha,$$

$$\rho^{1-\phi}Q'_\alpha = b_2 E_\alpha Q_\alpha - (\eta + d + c_2)Q_\alpha,$$

$$\rho^{1-\phi}U'_\alpha = b_1 E_\alpha U_\alpha - (w + d + c_3)U_\alpha, \tag{9}$$

$$\rho^{1-\phi}V'_\alpha = a_2 S_\alpha - (\sigma E_\alpha + d)V_\alpha,$$

$$\rho^{1-\phi}R'_\alpha = \eta Q_\alpha + wU_\alpha - dR_\alpha,$$

with initial conditions,

$$S_\alpha(0) \geq 0, \; E_\alpha(0) \geq 0, \; Q_\alpha(0) \geq 0, \; U_\alpha(0) \geq 0, \; V_\alpha(0) \geq 0, \; R_\alpha(0) \geq 0. \tag{10}$$

## 3 Model analysis

In this section, we will comprehensively discuss the differential analysis of the system, including the invariant region, positivity of solution, disease-free equilibrium point, basic reproduction number, sensitivity analysis, local and global stability at the disease-free equilibrium point, and the existence and uniqueness of the system.

### 3.1 Invariant region

To find the invariant region of system of equations (9) with non-negative initial conditions (10) solution is bounded, taking total population

$$\mathbb{N}(S_\alpha, E_\alpha, Q_\alpha, U_\alpha, V_\alpha, R_\alpha) = (S_\alpha(\rho) + E_\alpha(\rho) + Q_\alpha(\rho) + U_\alpha(\rho) + V_\alpha(\rho) + R_\alpha(\rho)).$$

In the absence of disease, take the derivative of $\mathbb{N}$ concerning $\rho$.

We obtain

$$\rho^{1-\phi}\mathbb{N}' = \beta - d\mathbb{N}, \tag{11}$$

after solving (11) and $\rho \to \infty$, then,

$$\Omega = \{(S_\alpha, E_\alpha, Q_\alpha, U_\alpha, V_\alpha, R_\alpha) \in \mathbb{R}^* : \mathbb{N}(t) \leq \frac{\beta}{d}\}, \tag{12}$$

which is the feasible solution set of a system of equations are bounded.

### 3.2 Positivity of solution

**Theorem 3.1**. *If*

$$S_\alpha(0) > 0, \; E_\alpha(0) > 0, \; Q_\alpha(0) > 0, \; U_\alpha(0) > 0, \; V_\alpha(0) > 0, \; R_\alpha(0) > 0$$

*are positive in the feasible set* $\Omega$, *then the solution set,*

$$(S_\alpha(\rho), \; E_\alpha(\rho), \; Q_\alpha(\rho), \; U_\alpha(\rho), \; V_\alpha(\rho), \; R_\alpha(\rho))$$

*of system of equations is positive* $\forall \rho \geq 0$.

*Proof.* Taking the first equation from the system of equations,

$$\rho^{1-\phi}S'_\alpha = \beta - (a_1 E_\alpha + a_2 + d)S_\alpha, \tag{13}$$

after simplification,

$$S \geq S(0)e^{-\rho^{\phi-1}(a_1 E_\alpha + a_2 + d)\rho}, \tag{14}$$

similar to another system of equations. Therefore, we can say the solution set of all systems of equations is positive for $\rho \geq 0$.

## 3.3 Disease-free equilibrium point (DFEP)

For the case, the population has no infectious individuals of COVID-19,

$$E_\alpha = Q_\alpha = U_\alpha = R_\alpha = 0.$$

Then disease-free equilibrium point is,

$$\mathbb{E}^0 = \begin{cases} S^0_\alpha = \dfrac{\beta}{a_2 + d}, \\[2mm] E^0_\alpha = 0, \\[2mm] Q^0_\alpha = 0, \\[2mm] U^0_\alpha = 0, \\[2mm] V^0_\alpha = \dfrac{\beta a_2}{d(a_2 + d)}, \\[2mm] R^0_\alpha = 0. \end{cases} \tag{15}$$

## 3.4 Basic reproduction number

The next-generation matrix method is used to calculate the basic reproduction number $R_0$ [31]. To determine $R_0$, we first derive the transmission matrix $\mathbb{A}$ from the system of equations (9) at the disease-free equilibrium point.

$$\mathbb{A} = \begin{pmatrix} \frac{a_1 \beta}{a_2 + d} + \frac{\sigma \beta a_2}{d(a_2 + d)} & 0 & 0 & 0 \\ 0 & 0 & 0 & 0 \\ 0 & 0 & 0 & 0 \\ 0 & 0 & 0 & 0 \end{pmatrix}. \tag{16}$$

Next, we derive the transition matrix $\mathbb{B}$ from the system of equations (9) at the disease-free equilibrium point.

$$\mathbb{B} = \begin{pmatrix} d + c_1 & 0 & 0 & 0 \\ 0 & -\eta - d - c_2 & 0 & 0 \\ 0 & 0 & -w - d - c_3 & 0 \\ 0 & -\eta & -w & d \end{pmatrix}. \tag{17}$$

We then compute the product $\mathbb{A}\mathbb{B}^{-1}$, which reflects the overall transmission potential considering both new infections and transitions between compartments:

$$
\mathbb{A}\mathbb{B}^{-1} = \begin{pmatrix} \dfrac{1}{d+c_1}\left(\dfrac{a_1\beta}{d+a_2} + \dfrac{\sigma\beta a_2}{d(d+a_2)}\right) & 0 & 0 & 0 \\[2ex] 0 & & 0 & 0 & 0 \\[2ex] 0 & & 0 & 0 & 0 \\[2ex] 0 & & 0 & 0 & 0 \end{pmatrix}. \tag{18}
$$

Finally, the basic reproduction number $R_0$ is derived from the dominant eigenvalue of $\mathbb{A}\mathbb{B}^{-1}$:

$$
R_0 = \frac{\beta(a_1 d + a_2 \sigma)}{d(da_2 + c_1 a_2 + d^2 + dc_1)}. \tag{19}
$$

This expression for $R_0$ provides insight into how various parameters affect the transmission dynamics of the infection. This shows that an increase in the transmission rate $\beta$ or the progression rates $a_1$ and $\sigma$ will raise $R_0$, indicating a higher potential for the disease to spread. Conversely, higher recovery or transition rates $d$ and $c_1$ can reduce $R_0$, highlighting the importance of timely interventions and effective disease management strategies, shown in Fig 3.

### 3.5 Sensitivity analysis

The factors contributing to this disease spread and persistence in the community are examined using sensitivity analysis. Our focus is on the variables that cause a greater variance in the basic reproduction number.

**Sensitivity indices of $R_0$.** The sensitivity index used to compute the corresponding variance in the state variable caused by the changing of a parameter. These indices have been calculated using the definition from [32]. The following definition of the sensitivity index is presented as partial derivatives:

$$
P_q^{R_0} = \frac{\partial R_0}{\partial q} \times \frac{q}{R_0}. \tag{20}
$$

The sensitivity indices of Eq (19) are given as follows,

$$
P_\beta^{R_0} = \frac{\partial R_0}{\partial \beta} \times \frac{\beta}{R_0} = 1 > 0, \tag{21}
$$

$$
P_{a_1}^{R_0} = \frac{\partial R_0}{\partial a_1} \times \frac{a_1}{R_0} = \frac{a_1 d}{a_2 \sigma + a_1 d} > 0, \tag{22}
$$

$$
P_{a_2}^{R_0} = \frac{\partial R_0}{\partial a_2} \times \frac{a_2}{R_0} = \frac{(\sigma - a_1) d a_2}{(d + a_2)(a_1 d + a_2 \sigma)} > 0, \tag{23}
$$

$$
P_{c_1}^{R_0} = \frac{\partial R_0}{\partial c_1} \times \frac{c_1}{R_0} = -\frac{c_1(d + a_2)}{d^2 + da_2 + dc_1 + c_1 a_2} < 0, \tag{24}
$$

$$
P_\sigma^{R_0} = \frac{\partial R_0}{\partial \sigma} \times \frac{\sigma}{R_0} = \frac{a_2 \sigma}{a_1 d + a_2 \sigma} > 0, \tag{25}
$$

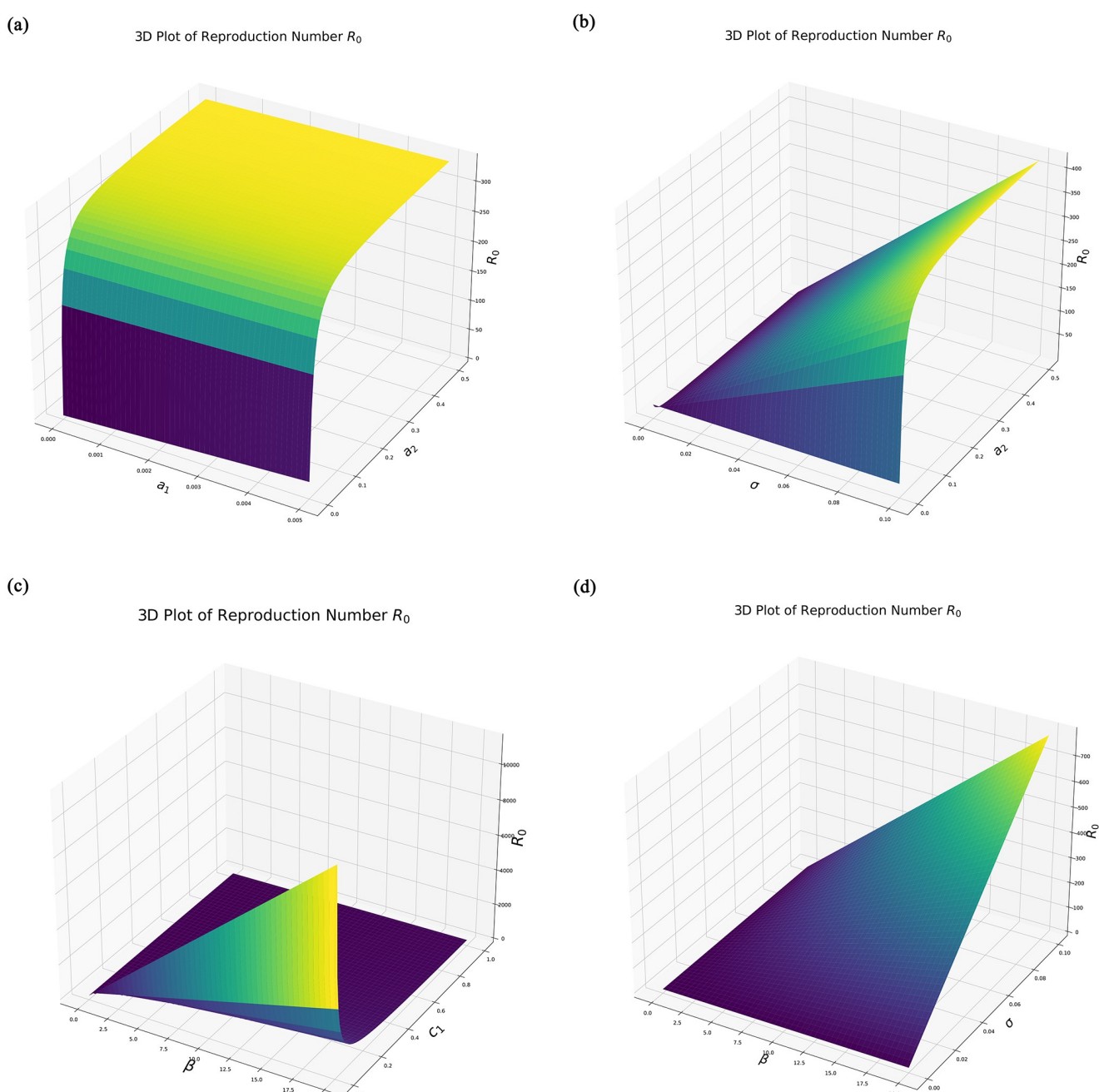

**Fig 3. Combined figure showing the behavior of $R_0$ in two different plots.** (A) Reproduction number $R_0$ between $a_1$ and $a_2$, (B) Reproduction number $R_0$ between $\sigma$ and $a_2$, (C) Reproduction number $R_0$ between $\beta$ and $c_1$, (D) Reproduction number $R_0$ between $\beta$ and $\sigma$.

$$P_d^{R_0} = \frac{\partial R_0}{\partial d} \times \frac{d}{R_0} = \frac{-2\,d^3 a_1 + ((-3\,\sigma - a_1)a_2 - a_1 c_1)d^2 - 2\,\sigma\,a_2(a_2 + c_1)d - \sigma\,a_2{}^2 c_1}{(a_1 d + a_2 \sigma)(d + c_1)(d + a_2)} < 0. \quad (26)$$

The explanation above demonstrates that the basic reproduction number $R_0$ is most sensitive to variations. If $\beta$ rises, $R_0$ will rise in proportionally the same way and $\beta$ if falls, $R_0$ will fall in proportionally the same way. The link between $c_1$, and $d$ is inversely proportional with $R_0$, therefore,

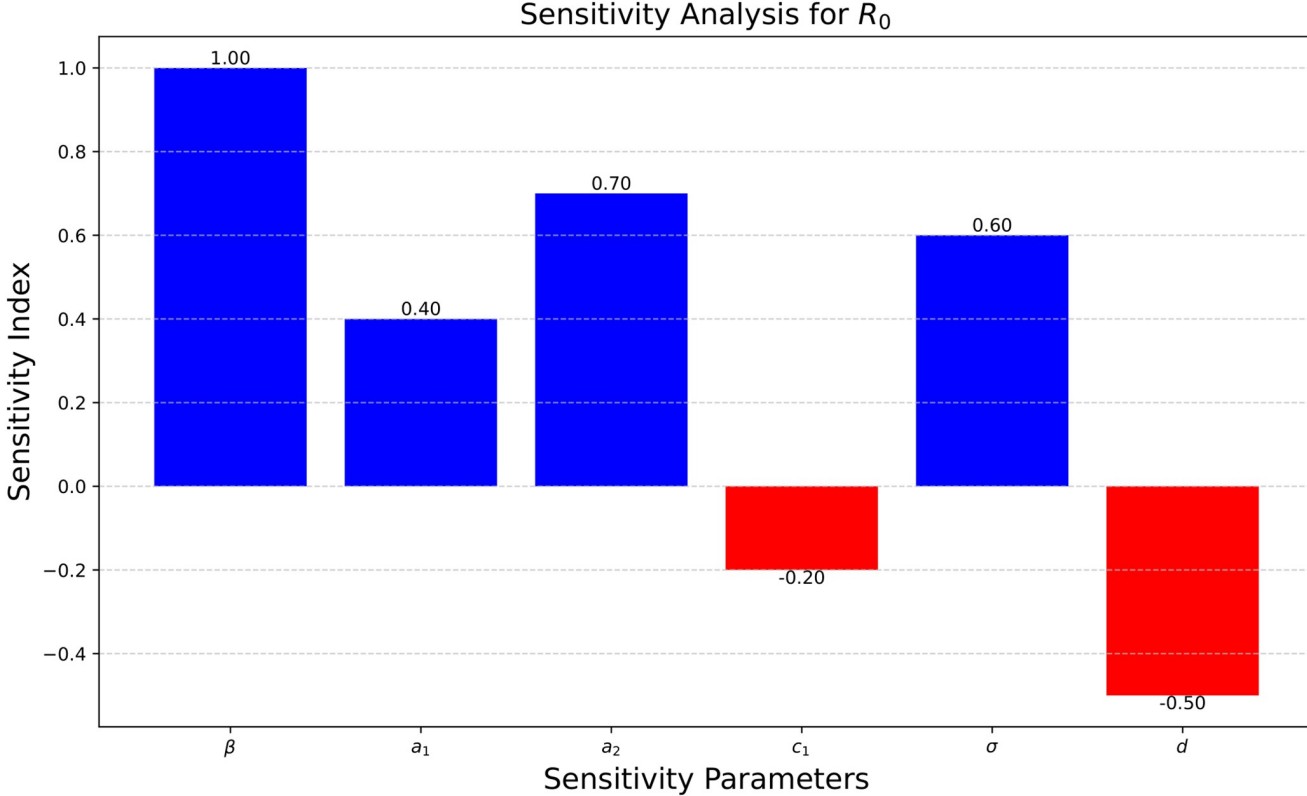

**Fig 4. Sensitivity analysis.**

an increase in either of these will result in a drop in $R_0$. It makes sense to concentrate on lowering these since, $a_1$, $a_2$, and $\sigma$ are more susceptible to changes than $R_0$ redshown in Fig 4. In other words, this sensitivity analysis shows us that preventing problems is preferable to fixing them. In addition to playing an important role in controlling the transmission of a virus, it is vital to remember that vaccinations also significantly contribute to reducing the impact of the disease.

### 3.6 Local stability of DFEP

**Theorem 3.2**. *The disease-free equilibrium point is locally stable if $R_0 < 1$ and unstable if $R_0 > 1$* [33].

*Proof*. The Jacobian matrix of system of equations (9) at disease-free equilibrium point,

$$J^* = \begin{pmatrix} -a_2 - d & -\dfrac{a_1\,\beta}{a_2 + d} & 0 & 0 & 0 & 0 \\[2mm] 0 & \dfrac{a_1\,\beta}{a_2 + d} + \dfrac{\sigma\,\beta\,a_2}{d(a_2 + d)} - d - c_1 & 0 & 0 & 0 & 0 \\[2mm] a_2 & -\dfrac{\sigma\,\beta\,a_2}{d(a_2 + d)} & -d & 0 & 0 & 0 \\[2mm] 0 & 0 & 0 & -d - \eta - c_2 & 0 & 0 \\[2mm] 0 & 0 & 0 & 0 & -d - w - c_3 & 0 \\[2mm] 0 & 0 & 0 & \eta & w & -d \end{pmatrix}. \quad (27)$$

The characteristic equations are,

$$\left(\frac{(a_1\,\beta\,d + \sigma\,\beta\,a_2 - a_2\,\chi\,d - a_2\,d^2 - a_2\,dc_1 - \chi\,d^2 - d^3 - d^2 c_1)}{d(a_2 + d)}\right) \times$$

$$\left(\frac{(-a_2 - d - \chi)(-d - \chi)^2(-d - \eta - c_2 - \chi)(-d - w - c_3 - \chi)}{d(a_2 + d)}\right), \tag{28}$$

where,

$$\begin{aligned}
\chi_1 &= -w - d - c_3, \chi_2 = -d, \chi_3 = -d, \chi_4 = -d - \eta - c_2, \\
\chi_5 &= -a_2 - d, \chi_6 = \frac{(da_2 + c_1 a_2 + d^2 + dc_1)}{a_2 + d}(R_0 - 1),
\end{aligned} \tag{29}$$

consist of solutions to the characteristic equation. We observe that $\chi_{1,2,3,4,5}$ have negative numbers, and if $R_0 < 1$, then it shows that $\chi_6$ is negative, indicating that DFEP is asymptotically stable locally. The proof is now completed.

### 3.7 Global stability

The concept proposed by Castillo-Chavez et al., [34] may be used to state the following results,

$$\begin{aligned}
\frac{dG_1}{dt} &= F(G_1, G_2), \\
\frac{dG_2}{dt} &= H(G_1, G_2),
\end{aligned} \tag{30}$$

where,

$$H(G_1, G_2) = 0. \tag{31}$$

When the system is at the DFEP, the infected and uninfected populations are represented by the values $G_1 \in \mathbb{R}^2$ and $G_2 \in \mathbb{R}^3$, respectively. The condition for global stability at the DFEP in the epidemiological model is given by:

$$\frac{dG_1}{dt} = F(G_1, 0) = 0, \tag{32}$$

$$H(G_1, G_2) = P_1 N_{G^*} - \hat{H}(G_1, G_2). \tag{33}$$

**Theorem 3.3**. *The system of equations* (9) *are globally asymptotically stable if* $R_0 < 1$ *at DFE point.*

*Proof.* To prove condition (32), the model (9) can be set by, the disease-free equilibrium point is given,

$$\mathbb{E}_0 = (X^0, 0) = \left(\frac{\beta}{a_2 + d}, \frac{a_2\,\beta}{d(a_2 + d)}\right), \tag{34}$$

and the system,

$$\frac{dG_1}{dt} = F(G_1, 0),$$ (35)

$$S_\alpha^* = \beta - (a_2 + d)S_\alpha,$$
$$V_\alpha^* = a_2 S_\alpha - (d)V_\alpha.$$ (36)

By solving Eq (36), the equation has a unique equilibrium point,

$$\left(S_\alpha^* = \frac{\beta}{a_2 + d}, V_\alpha^* = \frac{a_2 \beta}{d(a_2 + d)}\right),$$ (37)

hence, for the condition (32) $X^0$ is globally asymptotically stable is satisfied. Now, to verify the second condition (33).

$$H(G_1, G_2) = P_1 N_{G^*} - \hat{H}(G_1, G_2),$$ (38)

and,

$$\hat{H}(G_1, G_2) \geq 0,$$ (39)

$$H(G_1, G_2) = \begin{pmatrix} a_1 E_\alpha S_\alpha + \sigma V_\alpha E_\alpha - (b_1 U_\alpha + b_2 Q_\alpha + d + c_1)E_\alpha \\ b_2 E_\alpha Q_\alpha - (\eta + d + c_2)Q_\alpha \\ b_1 E_\alpha U_\alpha - (w + d + c_3)U_\alpha \\ -dR_\alpha + \eta Q_\alpha + wU_\alpha \end{pmatrix},$$ (40)

$$N_G^* = \begin{pmatrix} S_\alpha^* a_1 + V_\alpha^* \sigma - b_1 U_\alpha - b_2 Q_\alpha - d - c_1 & -b_2 E_\alpha & -b_1 E_\alpha & 0 \\ b_2 Q_\alpha & b_2 E_\alpha - d - \eta - c_2 & 0 & 0 \\ b_1 U_\alpha & 0 & b_1 E_\alpha - d - w - c_3 & 0 \\ 0 & \eta & w & -d \end{pmatrix},$$ (41)

$$\hat{H}(G_1, G_2) = \begin{pmatrix} ((V_\alpha^* - E_\alpha)\sigma + (S_\alpha^* - S_\alpha)a_1 - b_2 Q_\alpha - b_1 U_\alpha)E_\alpha \\ b_2 E_\alpha Q_\alpha \\ b_1 E_\alpha U_\alpha \\ 0 \end{pmatrix},$$ (42)

this shows that,

$$\hat{H}(G_1, G_2) \geq 0.$$ (43)

This result proves the conditions (32) and (33), indicating that the system is globally asymptotically stable when $R_0 < l$ at the DFEP. This completes the proof of Theorem 3.3.

### 3.8 Existence and uniqueness of solution

In differential calculus, the existence and uniqueness of solutions are crucial, as emphasized in numerous studies in [35–37]. In this section, we demonstrate the existence of solutions for the non-linear system of equations in the COVID-19 model (9) using fixed point theory with results proved in [35].

For non-linear system,

$$S_\alpha(\rho) - S_\alpha(0) = \int_\rho^\phi [\beta - (a_1 E_\alpha + a_2 + d)S_\alpha]d\rho,$$

$$E_\alpha(\rho) - E_\alpha(0) = \int_\rho^\phi [a_1 E_\alpha S_\alpha + \sigma V_\alpha E_\alpha - (b_1 U_\alpha + b_2 Q_\alpha + d + c_1)E_\alpha]d\rho,$$

$$Q_\alpha(\rho) - Q_\alpha(0) = \int_\rho^\phi [b_2 E_\alpha Q_\alpha - (\eta + d + c_2)Q_\alpha]d\rho,$$

$$U_\alpha(\rho) - U_\alpha(0) = \int_\rho^\phi [b_1 E_\alpha U_\alpha - (w + d + c_3)U_\alpha]d\rho, \tag{44}$$

$$V_\alpha(\rho) - V_\alpha(0) = \int_\rho^\phi [a_2 S_\alpha - (\sigma E_\alpha + d)V_\alpha]d\rho,$$

$$R_\alpha(\rho) - R_\alpha(0) = \int_\rho^\phi [\eta Q_\alpha + wU_\alpha - dR_\alpha]d\rho.$$

Now, let's start the procedure

$$S_\alpha(\rho) - S_\alpha(0) = \int_0^\rho \xi^{1-\phi}[\beta - (a_1 E_\alpha + a_2 + d)S_\alpha]d\phi,$$

$$E_\alpha(\rho) - E_\alpha(0) = \int_0^\rho \xi^{1-\phi}[a_1 E_\alpha S_\alpha + \sigma V_\alpha E_\alpha - (b_1 U_\alpha + b_2 Q_\alpha + d + c_1)E_\alpha]d\phi,$$

$$Q_\alpha(\rho) - Q_\alpha(0) = \int_0^\rho \xi^{1-\phi}[b_2 E_\alpha Q_\alpha - (\eta + d + c_2)Q_\alpha]d\phi,$$

$$U_\alpha(\rho) - U_\alpha(0) = \int_0^\rho \xi^{1-\phi}[b_1 E_\alpha U_\alpha - (w + d + c_3)U_\alpha]d\phi, \tag{45}$$

$$V_\alpha(\rho) - V_\alpha(0) = \int_0^\rho \xi^{1-\phi}[a_2 S_\alpha - (\sigma E_\alpha + d)V_\alpha]d\phi,$$

$$R_\alpha(\rho) - R_\alpha(0) = \int_0^\rho \xi^{1-\phi}[\eta Q_\alpha + wU_\alpha - dR_\alpha]d\phi.$$

Now, we define the kernels

$$\begin{aligned}
\Lambda_1(\rho, S_\alpha) &= \beta - (a_1 E_\alpha + a_2 + d)S_\alpha, \\
\Lambda_2(\rho, E_\alpha) &= a_1 E_\alpha S_\alpha + \sigma V_\alpha E_\alpha - (b_1 U_\alpha + b_2 Q_\alpha + d + c_1)E_\alpha, \\
\Lambda_3(\rho, Q_\alpha) &= b_2 E_\alpha Q_\alpha - (\eta + d + c_2)Q_\alpha, \\
\Lambda_4(\rho, U_\alpha) &= b_1 E_\alpha U_\alpha - (w + d + c_3)U_\alpha, \\
\Lambda_5(\rho, V_\alpha) &= a_2 S_\alpha - (\sigma E_\alpha + d)V_\alpha, \\
\Lambda_6(\rho, R_\alpha) &= \eta Q_\alpha + wU_\alpha - dR_\alpha.
\end{aligned} \tag{46}$$

**Theorem 3.4**. *If the following inequality in proven, then the kernels $\Lambda_1$, $\Lambda_2$, $\Lambda_3$, $\Lambda_4$, $\Lambda_5$ and $\Lambda_6$ satisfy the Lipschitz assumptions and contractions.*

$$0 \leq p_1, p_2, p_3, p_4, p_5, p_6 < 1, \tag{47}$$

*where $\|S_\alpha\| \leq k_1, \| E_\alpha \| \leq k_2, \| Q_\alpha \| \leq k_3, \| U_\alpha \| \leq k_4, \| V_\alpha \| \leq k_5, \| R_\alpha \| \leq k_6, p_1 = a_1 k_2 + a_2 + d, p_2 = a_1 k_1 + \sigma k_4 + b_2 k_3 + b_1 k_4 + b_2 k_3, p_3 = b_2 k_2, p_4 = b_1 k_2, p_5 = \sigma k_2, p_6 = d.$*

*Proof.* Consider $S_{\alpha 1}$ and $S_{\alpha 2}$ are two functions for the kernel $\Lambda_1$, then

$$\| \Lambda_1(\rho, S_{\alpha 1}) - \Lambda_1(\rho, S_{\alpha 2}) \| \quad \leq (a_1 b_2 + a_2 + d) \| S_1(\xi) - S_2(\xi) \|,$$
$$= p_1 \| S_{\alpha 1}(\rho) - S_{\alpha 2}(\rho) \|, \tag{48}$$

$k_1 = \|S\|$ is bounded function of $p_1$ then,

$$\| \Lambda_1(\rho, S_{\alpha 1}) - \Lambda_1(\rho, S_{\alpha 2}) \| \leq p_2 \| S_{\alpha 1}(\rho) - S_{\alpha 2}(\rho) \|, \tag{49}$$

when $E_1$ and $E_2$ are two bounded functions for the kernel $\lambda_2$, then similarly,

$$\| \Lambda_2(\rho, E_{\alpha 1}) - \Lambda_2(\rho, E_{\alpha 2}) \| \leq p_2 \| E_{\alpha 1}(\rho) - E_{\alpha 2}(\rho) \|, \tag{50}$$

when $Q_1$ and $Q_2$ are two bounded functions for the kernel $\lambda_3$, then similarly,

$$\| \Lambda_3(\rho, Q_{\alpha 1}) - \Lambda_3(\rho, Q_{\alpha 2}) \| \leq p_3 \| Q_{\alpha 1}(\rho) - Q_{\alpha 2}(\rho) \|, \tag{51}$$

when $U_1$ and $U_2$ are two bounded functions for the kernel $\Lambda_4$, then similarly,

$$\| \Lambda_4(\rho, U_{\alpha 1}) - \Lambda_4(\rho, U_{\alpha 2}) \| \leq p_4 \| U_{\alpha 1}(\rho) - U_{\alpha 2}(\rho) \|, \tag{52}$$

when $V_1$ and $V_2$ are two bounded function for the kernel $\Lambda_5$, then similarly,

$$\| \Lambda_5(\rho, V_{\alpha 1}) - \Lambda_5(\rho, V_{\alpha 2}) \| \leq p_5 \| V_{\alpha 1}(\rho) - V_{\alpha 2}(\rho) \|, \tag{53}$$

when $R_1$ and $R_2$ are two bounded functions for the kernel $\Lambda_6$, then similarly,

$$\| \Lambda_6(\rho, R_{\alpha 1}) - \Lambda_6(\rho, R_{\alpha 2}) \| \leq p_6 \| R_{\alpha 1}(\rho) - R_{\alpha 2}(\rho) \|, \tag{54}$$

therefore the $\Lambda_1$, $\Lambda_2$, $\Lambda_3$, $\Lambda_4$, $\Lambda_5$, $\Lambda_6$ satisfy the Lipschitz conditions.

If $0 \leq p_1, p_2, p_3 p_4, p_5, p_6 < 1$, then $p_1, p_2, p_3, p_4, p_5$ and $p_6$ also contraction for $\Lambda_1$, $\Lambda_2$, $\Lambda_3$, $\Lambda_4$, $\Lambda_5$, $\Lambda_6$ respectively. This is the proof of this theorem.

Now consider the kernels $\Lambda_1$, $\Lambda_2$, $\Lambda_3$, $\Lambda_4$, $\Lambda_5$, $\Lambda_6$ and rewrite the system of equations,

$$S_\alpha(\rho) = S_\alpha(0) + \int_0^\rho \Lambda_1(\sigma, S_\alpha) d\xi$$

$$E_\alpha(\rho) = E_\alpha(0) + \int_0^\rho \Lambda_2(\sigma, E_\alpha) d\xi,$$

$$Q_\alpha(\rho) = Q_\alpha(0) + \int_0^\rho \Lambda_3(\sigma, Q_\alpha) d\xi,$$

$$U_\alpha(\rho) = U_\alpha(0) + \int_0^\rho \Lambda_4(\sigma, U_\alpha) d\xi, \tag{55}$$

$$V_\alpha(\rho) = V_\alpha(0) + \int_0^\rho \Lambda_5(\sigma, V_\alpha) d\xi,$$

$$R_\alpha(\rho) = R_\alpha(0) + \int_0^\rho \Lambda_6(\sigma, R_\alpha) d\xi.$$

Now proceed with the recursive formula, which is as follows,

$$S_{\alpha r}(\rho) = S_\alpha(0) + \int_0^\rho \Lambda_1(\sigma, S_{\alpha(r-1)}) d\xi,$$

$$E_{\alpha r}(\rho) = E_\alpha(0) + \int_0^\rho \Lambda_2(\sigma, E_{\alpha(r-1)}) d\xi,$$

$$Q_{\alpha r}(\rho) = Q_\alpha(0) + \int_0^\rho \Lambda_3(\sigma, Q_{\alpha(r-1)}) d\xi,$$

$$U_{\alpha r}(\rho) = U_\alpha(0) + \int_0^\rho \Lambda_4(\sigma, U_{\alpha(r-1)}) d\xi,$$

$$V_{\alpha r}(\rho) = V_\alpha(0) + \int_0^\rho \Lambda_5(\sigma, V_{\alpha(r-1)}) d\xi,$$

$$R_{\alpha r}(\rho) = R_\alpha(0) + \int_0^\rho \Lambda_7(\sigma, R_{\alpha(r-1)}) d\xi,$$

(56)

where,

$$S_\alpha(0) \geq 0, \ E_\alpha(0) \geq 0, \ Q_\alpha(0) \geq 0, \ U_\alpha(0) \geq 0, \ V_\alpha(0) \geq 0, \ R_\alpha(0) \geq 0. \tag{57}$$

It can also be written in sequential term differences which are as follows,

$$\omega 1_r = S_\alpha(\xi) - S_\alpha(0) = \int_0^\rho \xi^{c-1}(\Lambda_1(\xi, S_{r-1}) - \Lambda_1(\xi, S_{r-2})) d\xi,$$

$$\omega 2_r = E_\alpha(\xi) - E_\alpha(0) = \int_0^\rho \xi^{c-1}(\Lambda_2(\xi, E_{r-1}) - \Lambda_2(\xi, E_{r-2})) d\xi,$$

$$\omega 3_r = Q_\alpha(\xi) - Q_\alpha(0) = \int_0^\rho \xi^{c-1}(\Lambda_3(\xi, Q_{r-1}) - \Lambda_3(\xi, Q_{r-2})) d\xi,$$

$$\omega 4_r = U_\alpha(\xi) - U_\alpha(0) = \int_0^\rho \xi^{c-1}(\Lambda_4(\xi, U_{r-1}) - \Lambda_4(\xi, U_{r-2})) d\xi,$$

$$\omega 5_r = V_\alpha(\xi) - V_\alpha(0) = \int_0^\rho \xi^{c-1}(\Lambda_5(\xi, J_{r-1}) - \Lambda_5(\xi, J_{r-2})) d\xi,$$

$$\omega 6_r = R_\alpha(\xi) - R_\alpha(0) = \int_0^\rho \xi^{c-1}(\Lambda_6(\xi, R_{r-1}) - \Lambda_6(\xi, R_{r-2})) d\xi,$$

(58)

this system of equations implies that,

$$S_{\alpha r}(\rho) = \sum_{j=1}^r \omega 1_r(\rho),$$

$$E_{\alpha r}(\rho) = \sum_{j=1}^r \omega 2_r(\rho),$$

$$Q_{\alpha r}(\rho) = \sum_{j=1}^r \omega 3_r(\rho),$$

$$U_{\alpha r}(\rho) = \sum_{j=1}^r \omega 4_r(\rho),$$

$$V_{\alpha r}(\rho) = \sum_{j=1}^r \omega 5_r(\rho),$$

$$R_{\alpha r}(\rho) = \sum_{j=1}^r \omega 5_r(\rho).$$

(59)

Now, we take both sides of the system of equations, then kernels satisfy the Lipschitz condition. Now triangle inequality applies to a system of equations, then we have,

$$\| S_{\alpha r}(\rho) - S_{\alpha(r-1)}(\rho) \| \le p_1 \int_0^\rho \xi^{\phi-1} \| (S_{\alpha(r-1)} - S_{\alpha(r-2)}) \| d\xi,$$

$$\| E_{\alpha r}(\rho) - E_{\alpha(r-1)}(\rho) \| \le p_2 \int_0^\rho \xi^{\phi-1} \| (E_{\alpha(r-1)} - E_{\alpha(r-2)}) \| d\xi,$$

$$\| Q_{\alpha r}(\rho) - Q_{\alpha(r-1)}(\rho) \| \le p_3 \int_0^\rho \xi^{\phi-1} \| (Q_{\alpha(r-1)} - Q_{\alpha(r-2)}) \| d\xi,$$

$$\| U_{\alpha r}(\rho) - U_{\alpha(r-1)}(\rho) \| \le p_4 \int_0^\rho \xi^{\phi-1} \| (U_{\alpha(r-1)} - U_{\alpha(r-2)}) \| d\xi,$$

$$\| V_{\alpha r}(\rho) - V_{\alpha(r-1)}(\rho) \| \le p_5 \int_0^\rho \xi^{\phi-1} \| (V_{\alpha(r-1)} - V_{\alpha(r-2)}) \| d\xi,$$

$$\| R_{\alpha r}(\rho) - R_{\alpha(r-1)}(\rho) \| \le p_6 \int_0^\rho \xi^{\phi-1} \| (R_{\alpha(r-1)} - R_{\alpha(r-2)}) \| d\xi,$$

$$(60)$$

we have,

$$\| \omega 1_r \| \le p_1 \int_0^\rho \| \omega 1_{r-1} \| d\xi, \| \omega 1_r \| \le p_2 \int_0^\rho \| \omega 1_{r-1} \| d\xi,$$

$$\| \omega 1_r \| \le p_3 \int_0^\rho \| \omega 1_{r-1} \| d\xi, \| \omega 1_r \| \le p_4 \int_0^\rho \| \omega 1_{r-1} \| d\xi,$$

$$\| \omega 1_r \| \le p_5 \int_0^\rho \| \omega 1_{r-1} \| d\xi, \| \omega 1_r \| \le p_6 \int_0^\rho \| \omega 1_{r-1} \| d\xi.$$

$$(61)$$

The following theorem may be derived from these findings.

**Theorem 3.5**. *The modified COVID-19 model offers a solution under the condition that can be formed $\tau_{max}$ property*,

$$p_i \tau_{max} \le 1, i = 1, 2, \cdots, 7. \tag{62}$$

*Proof*. Consider the function $S_\alpha(\rho)$, $E_\alpha(\rho)$, $Q_\alpha(\rho)$, $U_\alpha(\rho)$, $V_\alpha(\rho)$ and $R_\alpha(\rho)$ are the bounded and having the kernels $\Lambda_1, \Lambda_2, \Lambda_3, \Lambda_4, \Lambda_5, \Lambda_6$ satisfied the Lipschitz condition. We apply the recursive method to a system of equations,

$$\| \omega 1_r \| \le S_\alpha(0) \| \{ p_1 \tau_{max} \}^r,$$

$$\| \omega 2_r \| \le E_\alpha(0) \| \{ p_2 \tau_{max} \}^r,$$

$$\| \omega 3_r \| \le Q_\alpha(0) \| \{ p_3 \tau_{max} \}^r,$$

$$\| \omega 4_r \| \le U_\alpha(0) \| \{ p_4 \tau_{max} \}^r,$$

$$\| \omega 5_r \| \le V_\alpha(0) \| \{ p_5 \tau_{max} \}^r,$$

$$\| \omega 6_r \| \le R_\alpha(0) \| \{ p_6 \tau_{max} \}^r,$$

$$(63)$$

so this is the solution of the COVID-19 model, we suppose that,

$$S_\alpha(\rho) - S_\alpha(0) = S_{\alpha r}(\rho) - Z1_n(\rho), E_\alpha(\rho) - E_\alpha(0) = E_{\alpha r}(\rho) - Z2_n(\rho),$$

$$Q_\alpha(\rho) - Q_\alpha(0) = Q_{\alpha r}(\rho) - Z3_n(\rho), U_\alpha(\rho) - U_\alpha(0) = U_{\alpha r}(\rho) - Z4_n(\rho), \quad (64)$$

$$V_\alpha(\rho) - V_\alpha(0) = V_{\alpha r}(\rho) - Z5_n(\rho), R_\alpha(\rho) - R_\alpha(0) = R_{\alpha r}(\rho) - Z6_n(\rho).$$

It is shown that the term in Eq (64) hold, $\|Z1_n(\rho)\| \to 0$, $\|Z2_n(\rho)\| \to 0$, $\|Z3_n(\rho)\| \to 0$, $\|Z4_n(\rho)\| \to 0$, $\|Z5_n(\rho)\| \to 0$, $\|Z6_n(\rho)\| \to 0$, so we have,

$$
\begin{aligned}
\| Z1_n(\rho) \| \quad &\leq \| \int_0^\rho \xi^{\phi-1}[\Lambda_1(\xi, S_\alpha) - \Lambda_1(\xi, S_{\alpha(r-1)})]d\rho \|, \\
&\leq \int_0^\rho \| \xi^{\phi-1}[\Lambda_1(\xi, S_\alpha) - \Lambda_1(\xi, S_{\alpha(r-1)})] \| d\xi, \\
&\leq \xi p_1 \| S_\alpha - S_{\alpha(r-1)} \| .
\end{aligned}
\quad (65)
$$

Similarly for others,

$$\| Z2_n(\rho) \| \leq p_2 \rho \| E_\alpha - E_{\alpha(r-1)} \|, \quad (66)$$

$$\| Z3_n(\rho) \| \leq p_3 \rho \| Q_\alpha - Q_{\alpha(r-1)} \|, \quad (67)$$

$$\| Z4_n(\rho) \| \leq p_4 \rho \| U_\alpha - U_{\alpha(r-1)} \|, \quad (68)$$

$$\| Z5_n(\rho) \| \leq p_5 \rho \| V_\alpha - V_{\alpha(r-1)} \|, \quad (69)$$

and,

$$\| Z6_n(\rho) \| \leq p_6 \rho \| R_\alpha - R_{\alpha(r-1)} \|, \quad (70)$$

apply recursive relation, then we obtain

$$
\begin{aligned}
\| Z1_n(\rho) \| &\leq \phi^{r-1} p_1^r \Phi, \\
\| Z2_n(\rho) \| &\leq \phi^{r-1} p_2^r \Phi, \\
\| Z3_n(\rho) \| &\leq \phi^{r-1} p_3^r \Phi, \\
\| Z4_n(\rho) \| &\leq \phi^{r-1} p_4^r \Phi, \\
\| Z5_n(\rho) \| &\leq \phi^{r-1} p_5^r \Phi, \\
\| Z6_n(\rho) \| &\leq \phi^{r-1} p_6^r \Phi,
\end{aligned}
\quad (71)
$$

taking at $\tau_{max}$ point, we get

$$
\begin{aligned}
\| Z1_n(\rho \| &\leq \{\tau_{max}\}^{r-1} p_1^r \Phi, \\
\| Z2_n(\rho \| &\leq \{\tau_{max}\}^{r-1} p_2^r \Phi, \\
\| Z3_n(\rho \| &\leq \{\tau_{max}\}^{r-1} p_3^r \Phi, \\
\| Z4_n(\rho \| &\leq \{\tau_{max}\}^{r-1} p_4^r \Phi, \\
\| Z5_n(\rho \| &\leq \{\tau_{max}\}^{r-1} p_5^r \Phi, \\
\| Z6_n(\rho \| &\leq \{\tau_{max}\}^{r-1} p_6^r \Phi,
\end{aligned}
\quad (72)
$$

as $r \to \infty$ apply both sides, then using the result of theorem 3.4, then we get,

$$\| Z1_n(\rho) \| \to 0,$$

$$\| Z2_n(\rho) \| \to 0,$$

$$\| Z3_n(\rho) \| \to 0,$$

$$\| Z4_n(\rho) \| \to 0,$$

$$\| Z5_n(\rho) \| \to 0,$$

$$\| Z6_n(\rho) \| \to 0.$$

**Theorem 3.6**. *if*

$$(1 - p_i\rho) \geq 0, \ i = 1, 2, \cdots, 6, \tag{73}$$

*then modified COVID-19 model has a unique system of solutions.*

*Proof.* Suppose different system of solution such as $\hat{S}_\alpha, \hat{E}_\alpha, \ \hat{Q}_\alpha, \ \hat{U}_\alpha, \ \hat{V}_\alpha, \ \hat{R}_\alpha$, then it may write,

$$
\begin{aligned}
S_\alpha(\rho) - \hat{S}_\alpha(\rho) &= \int_0^\rho \xi^{\phi-1}[\Lambda_1(\xi, S_\alpha) - \Lambda_1(\xi, \hat{S}_\alpha)]d\xi, \\
E_\alpha(\rho) - \hat{E}_\alpha(\rho) &= \int_0^\rho \xi^{\phi-1}[\Lambda_2(\xi, E_\alpha) - \Lambda_2(\xi, \hat{E}_\alpha)]d\xi, \\
Q_\alpha(\rho) - \hat{Q}_\alpha(\rho) &= \int_0^\rho \xi^{\phi-1}[\Lambda_3(\xi, Q_\alpha) - \Lambda_3(\xi, \hat{Q}_\alpha)]d\xi, \\
U_\alpha(\rho) - \hat{U}_\alpha(\rho) &= \int_0^\rho \xi^{\phi-1}[\Lambda_4(\xi, U_\alpha) - \Lambda_4(\xi, \hat{U}_\alpha)]d\xi, \\
V_\alpha(\rho) - \hat{V}_\alpha(\rho) &= \int_0^\rho \xi^{\phi-1}[\Lambda_5(\xi, V_\alpha) - \Lambda_5(\xi, \hat{V}_\alpha)]d\xi, \\
R_\alpha(\rho) - \hat{R}_\alpha(\rho) &= \int_0^\rho \xi^{\phi-1}[\Lambda_6(\xi, R_\alpha) - \Lambda_6(\xi, \hat{R}_\alpha)]d\xi.
\end{aligned}
\tag{74}
$$

Apply norm on both sides (74) and results of kernels which fulfil the Lipschitz condition. We can write it as,

$$
\begin{aligned}
\| S_\alpha(\rho) - \hat{S}_\alpha(\rho) \| &\leq p_1\xi \| S_\alpha(\rho) - \hat{S}_\alpha(\rho) \|, \\
\| E_\alpha(\rho) - \hat{E}_\alpha(\rho) \| &\leq p_2\xi \| E_\alpha(\rho) - \hat{E}_\alpha(\rho) \|, \\
\| Q_\alpha(\rho) - \hat{S}_\alpha(\rho) \| &\leq p_3\xi \| Q_\alpha(\rho) - \hat{Q}_\alpha(\rho) \|, \\
\| U_\alpha(\rho) - \hat{U}_\alpha(\rho) \| &\leq p_4\xi \| U_\alpha(\rho) - \hat{U}_\alpha(\rho) \|, \\
\| V_\alpha(\rho) - \hat{V}_\alpha(\rho) \| &\leq p_5\xi \| V_\alpha(\rho) - \hat{V}_\alpha(\rho) \|, \\
\| R_\alpha(\rho) - \hat{R}_\alpha(\rho) \| &\leq p_6\xi \| R_\alpha(\rho) - \hat{R}_\alpha(\rho) \|.
\end{aligned}
\tag{75}
$$

then,

$$\| S_\alpha(\rho) - \hat{S}_\alpha(\rho) \| (1 - p_1\rho) \leq 0,$$
$$\| E_\alpha(\rho) - \hat{E}_\alpha(\rho) \| (1 - p_2\rho) \leq 0,$$
$$\| Q_\alpha(\rho) - \hat{Q}_\alpha(\rho) \| (1 - p_3\rho) \leq 0,$$
$$\| U_\alpha(\rho) - \hat{U}_\alpha(\rho) \| (1 - p_4\rho) \leq 0, \tag{76}$$
$$\| V_\alpha(\rho) - \hat{V}_\alpha(\rho) \| (1 - p_5\rho) \leq 0,$$
$$\| R_\alpha(\rho) - \hat{R}_\alpha(\rho) \| (1 - p_6\rho) \leq 0,$$

consequently,

$$\| S_\alpha(\rho) - \hat{S}_\alpha(\rho) \| = 0,$$
$$\| E_\alpha(\rho) - \hat{E}_\alpha(\rho) \| = 0,$$
$$\| Q_\alpha(\rho) - \hat{Q}_\alpha(\rho) \| = 0,$$
$$\| U_\alpha(\rho) - \hat{U}_\alpha(\rho) \| = 0, \tag{77}$$
$$\| V_\alpha(\rho) - \hat{V}_\alpha(\rho) \| = 0,$$
$$\| R_\alpha(\rho) - \hat{R}_\alpha(\rho) \| = 0.$$

This shows that the model has a unique solution. which is the complete proof of the theorem.

## 4 Numerical simulations

In this section, we explore the numerical results for the model we developed, considering various vaccination strategies. Furthermore, the essential assumption of the FDM is based on Taylor's theorem proved from [38], which asserts the following:

$$S(\rho + y) = S(\rho) + yS'(t) + \frac{y^2}{2!}S''(\rho) + \cdots, \tag{78}$$

For the fractional case, we have:

$$S(\rho + y) = S(\rho) + \sum_{k=1}^{\infty} \frac{y^k}{k!} \mathbb{D}^k S(\rho). \tag{79}$$

Where $\mathbb{D}^k$ shows the $k^{th}$ order derivative. For conformable fractional derivatives, we have

$$S(\rho + y) = S(\rho) + \sum_{k=1}^{\infty} \frac{\phi}{h^\phi} \mathbb{D}^{k\phi} S(\rho), \ 0 < \phi \leq 1. \tag{80}$$

### 4.1 FDM approximations for conformable fractional derivatives

Using the fractional order approximations for the forward finite difference method as discussed in [39, 40] and based on the results mentioned in Eq (80), we derive the following:

$$\mathbb{D}^{k\phi} S(\rho_n) = \frac{1}{\Delta\rho}(\phi)^\phi \sum_{i=0}^{k} (-1)^i \binom{k}{i} S(n+k-i), \quad 0 < \phi \leq 1, k \in N. \tag{81}$$

**Theorem 4.1**. *The relationship between classical and fractional order derivatives using the finite difference method can be expressed by the generalized recurrence relation presented in*

Eq (81),

$$\mathbb{D}^{\phi k} = (\Delta \rho)^{k(1-\phi)}(k\phi)\mathbb{D}^k, 0 < \phi \leq 1. \wedge k \in N. \tag{82}$$

**Proof**:

$$\frac{dS}{d\rho} = \frac{S_{n+1} - S_n}{\Delta \rho}. \tag{83}$$

Using fractional order Taylor series,

$$\frac{d^\phi S}{St^\phi} = \frac{S_{n+1} - S_n}{(\Delta t)^\phi}(\phi), 0 < \phi \leq 1. \tag{84}$$

From Eq (83), we get

$$u_i^{n+1} - u_i^n = (\Delta t)\frac{\partial u}{\partial t}. \tag{85}$$

From Eq (84), we get

$$S_{n+1} - S_n = \frac{(\Delta \rho)^\phi}{\phi}\frac{d^\phi S}{d\rho^\phi}. \tag{86}$$

Comparing Eqs (85) and (86), we get

$$(\Delta \rho)\frac{dS}{d\rho} = \frac{(\Delta \rho)^\phi}{\phi}\frac{d^\phi S}{d\rho^\phi}.$$

After rearranging, we get

$$\frac{d^\phi S}{d\rho^\phi} = \frac{\phi}{(\Delta \rho)^{\phi-1}}\frac{dS}{d\rho}, 0 < \phi \leq 1, \tag{87}$$

or also can be written as,

$$\frac{d^\phi S}{d\rho^\phi} = (\Delta \rho)^{1-\phi}\phi\frac{dS}{d\rho}, 0 < \phi \leq 1. \tag{88}$$

Applying this method to the system of equations (9), we obtain:

$$(\Delta \rho)^{1-\phi}\phi\frac{S_{\alpha(n+1)} - S_{\alpha(n)}}{\Delta(\rho)} = \beta - \left(a_1 E_{\alpha(n)} + a_2 + d\right)S_{\alpha(n)},$$

$$(\Delta \rho)^{1-\phi}\phi\frac{E_{\alpha(n+1)} - E_{\alpha(n)}}{\Delta(\rho)} = a_1 E_\alpha S_{\alpha(n)} + \sigma V_{\alpha(n)} E_{\alpha(n)}$$

$$-(b_1 U_{\alpha(n)} + b_2 Q_\alpha + d + c_1)E_{\alpha(n)},$$

$$(\Delta \rho)^{1-\phi}\phi\frac{Q_{\alpha(n+1)} - Q_{\alpha(n)}}{\Delta(\rho)} = b_2 E_{\alpha(n)}Q_{\alpha(n)} - (\eta + d + c_2)Q_{\alpha(n)}, \tag{89}$$

$$(\Delta \rho)^{1-\phi}\phi\frac{U_{\alpha(n+1)} - U_{\alpha(n)}}{\Delta(\rho)} = b_1 E_{\alpha(n)}U_{\alpha(n)} - (w + d + c_3)U_{\alpha(n)},$$

$$(\Delta \rho)^{1-\phi}\phi\frac{V_{\alpha(n+1)} - V_{\alpha(n)}}{\Delta(\rho)} = a_2 S_{\alpha(n)} - \left(\sigma E_{\alpha(n)} + d\right)V_{\alpha(n)},$$

$$(\Delta \rho)^{1-\phi}\phi\frac{R_{\alpha(n+1)} - R_{\alpha(n)}}{\Delta(\rho)} = \eta Q_{\alpha(n)} + wU_{\alpha(n)} - dR_{\alpha(n)}.$$

**Table 2. Values of parameters.**

| Parameters | Values | Sources |
|---|---|---|
| $a_1$ | 0.002 | [11] |
| $a_2$ | 0.5 | [11] |
| $\beta$ | 50 | [11] |
| $b_1$ | 0.0786 | Assumed |
| $b_2$ | 0.008 | [11] |
| $d$ | 0.009 | [11] |
| $c$ | 0.25 | [11] |
| $\sigma$ | 0.08 | [11] |
| $\eta$ | 0.5 | [11] |
| $w$ | 0.02 | Assumed |
| $S_\alpha$ | 50 | Assumed |
| $E_\alpha$ | 20 | [11] |
| $U_\alpha$ | 10 | [11] |
| $Q_\alpha$ | 6 | Assumed |
| $V_\alpha$ | 0 | [11] |
| $R_\alpha$ | 0 | [11] |

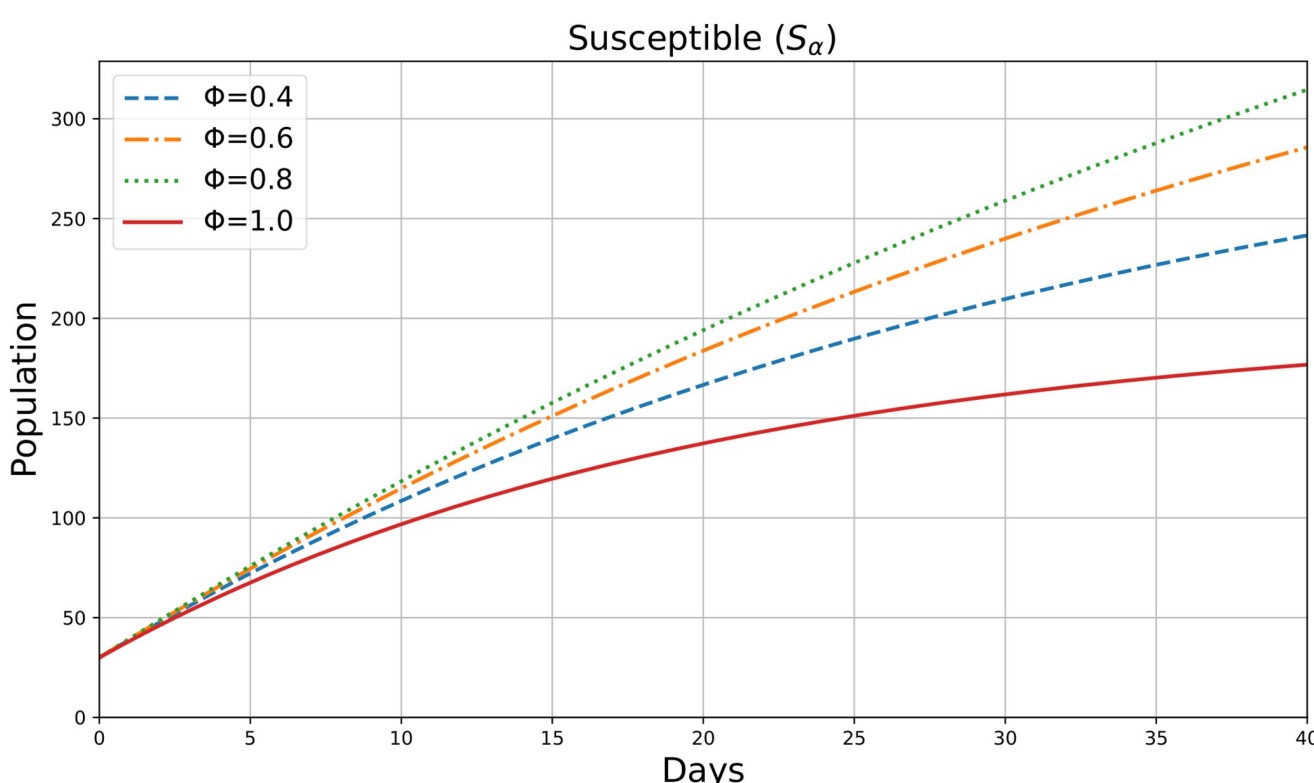

**Fig 5. Susceptible individuals.**

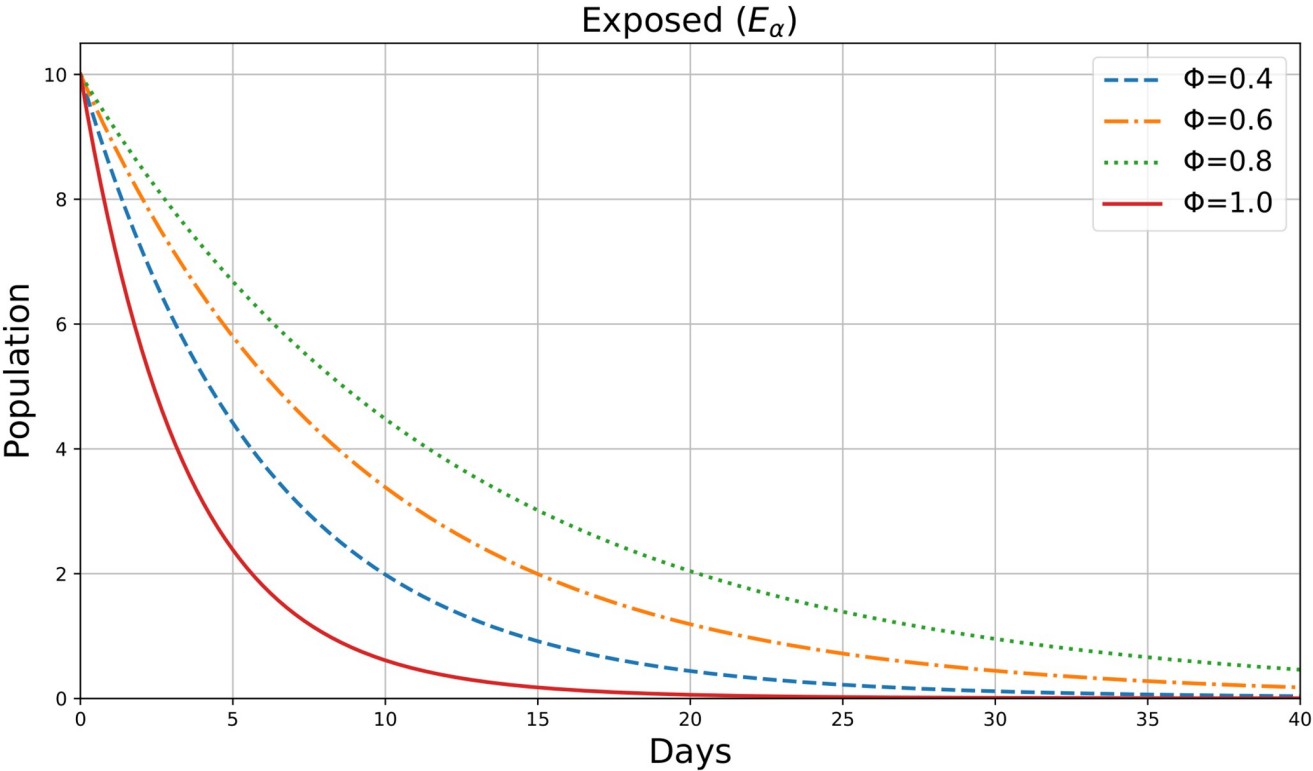

**Fig 6. Exposed individuals.**

Now, to establish the stability, consistency, and convergence of the system, the analysis will be based on the novel scheme.

## 4.2 Stability analysis

According to [41], in order to establish the stability of first equation from system (89), we initially simplify the analysis by neglecting certain factors,

$$(\Delta\rho)^{1-\phi}\phi\frac{S_{\alpha(n+1)} - S_{\alpha(n)}}{\Delta(\rho)} = -\left(a_1 E_{\alpha(n)} + a_2 + d\right)S_{\alpha(n)}. \tag{90}$$

After simplification, we have

$$R_1[S_{\alpha(n+1)}] = -[R_1 + R_2]S_{\alpha(n)}.$$

Where $R_1 = \frac{(\delta\rho)^{1-\phi}\phi}{\delta\rho}$ and $R_2 = a_1 E_{\alpha(n)} + a_2 + d$. Then, Equation (4.2) yields:

$$\frac{S_{\alpha(n+1)}}{S_{\alpha(n)}} = \frac{-[R_1 + R_2]}{R_1} < 1.$$

This shows that first equation of system (89) is stable. Similarly, we can prove other equations of system (89).

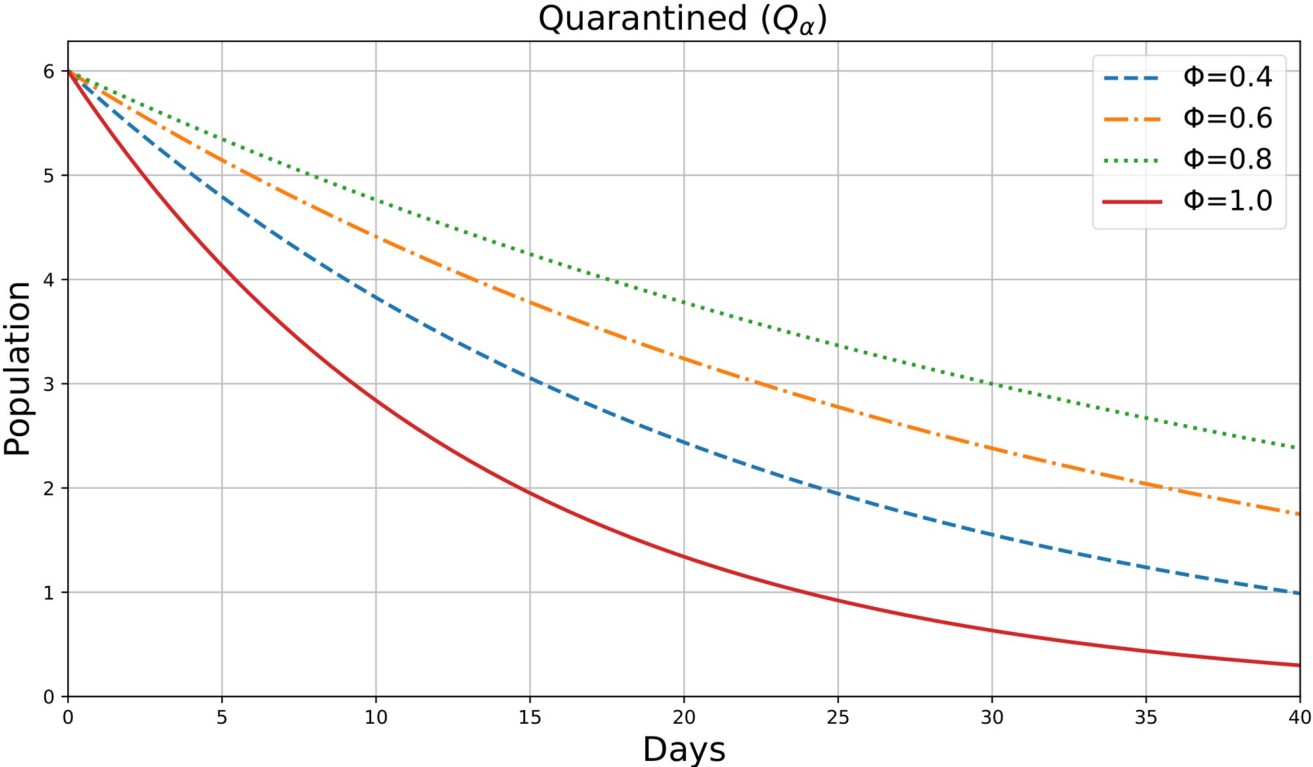

**Fig 7. Quarantined individuals.**

### 4.3 Consistency

As the grid interval and time step size approaches zero, the truncation error vanishes. Consistency, as discussed in [41], assesses the accuracy with which the finite difference method approximates the ordinary differential equation (ODE). We consider first equation from system (9) to be:

$$\rho^{1-\phi}\frac{dS_\alpha}{d\rho} = \beta - (a_1 E_\alpha + a_2 + d)S_\alpha, \tag{91}$$

Using the Taylor series, we have

$$S_{\alpha(n+1)} = S_{\alpha(n)} + (\Delta\rho)\frac{dS_\alpha}{d\rho} + \frac{(\Delta t)^2}{2!}\frac{d^2 S_\alpha}{d\rho^2} + \frac{(\Delta\rho)^3}{3!}\frac{d^3 S_\alpha}{d\rho^3} + \dots$$

After Substitution in Eq (91), we have

$$\rho^{1-\phi}\left[\frac{dS_\alpha}{d\rho} + \frac{\Delta\rho}{2!}\frac{d^2 S_\alpha}{d\rho^2} + O(\Delta\rho^2)\right] = \beta - (a_1 E_\alpha + a_2 + d)S_\alpha. \tag{92}$$

Therefore, the truncation error for the Eq (91) is as follows:

$$\tau = \frac{\Delta\rho}{2!}D^\phi S_\alpha(t) + O(\Delta\rho^2).$$

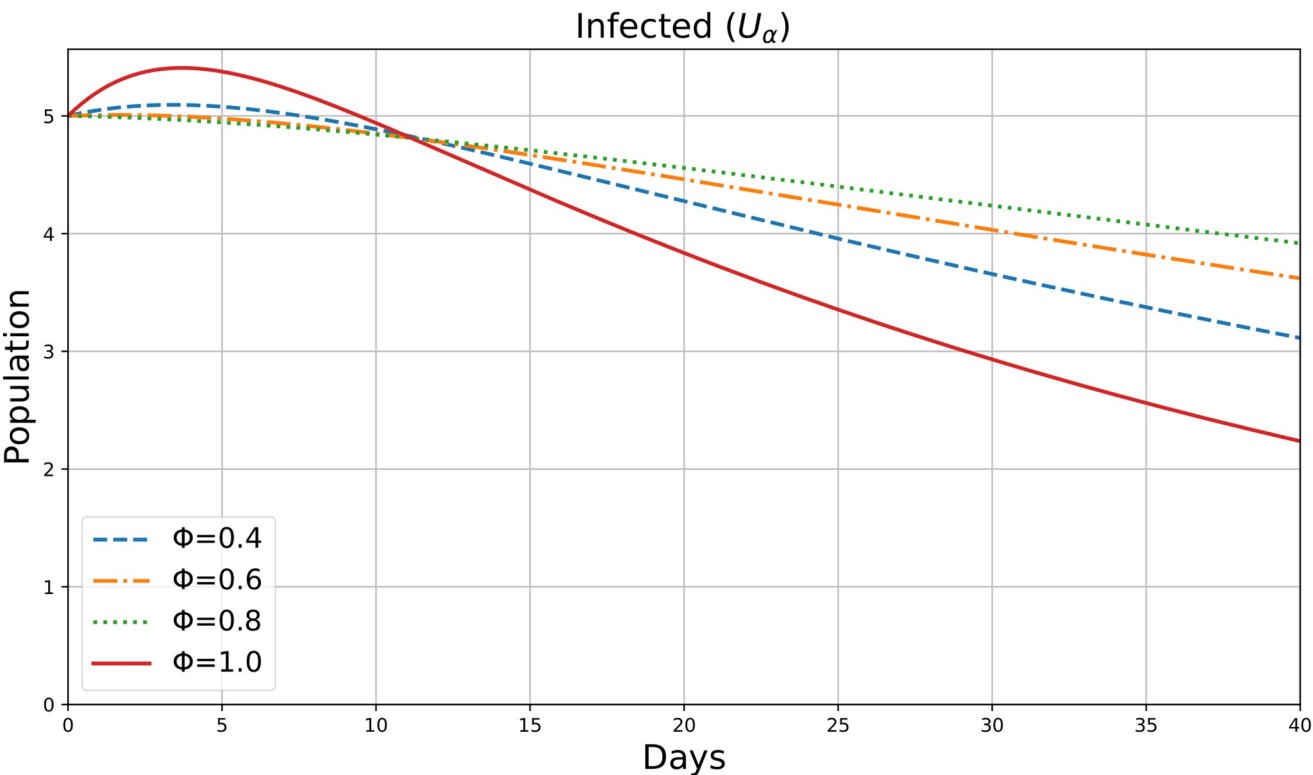

**Fig 8. Infected individuals.**

The truncation error $\tau \to 0$ when $\Delta\rho \to 0$. Similarly, we can prove other equations of system (89).

## 4.4 Convergence

Using the Lax-Richtmyer Equivalence theorem mentioned in [42],

$$Stability + Consistency \Leftrightarrow Convergence.$$

Then governing equations of the mathematical model are convergent.

## 4.5 Graphical behavior

Now, we have plot the system of equations (89) using the parameters values mentioned in Table 2.

This figure illustrates the plot of the variable $S_\alpha(\rho)$, which likely represents the number of susceptible individuals in society over time $\rho$ (days), for different values of the parameter $\phi = 0.4, 0.6, 0.8,$ and 1. Fig 5 shows the dynamic change in the number of susceptible individuals over a period of 40 days. The results indicate that the number of susceptible individuals is increasing, suggesting a decreasing chance of infection. As $\phi$ increases, this decrease in the infection rate becomes more pronounced. For lower values of $\phi$, the decrease is slower, indicating that the influence of fractional derivatives significantly affects the transition rate from susceptible to exposed or other compartments.

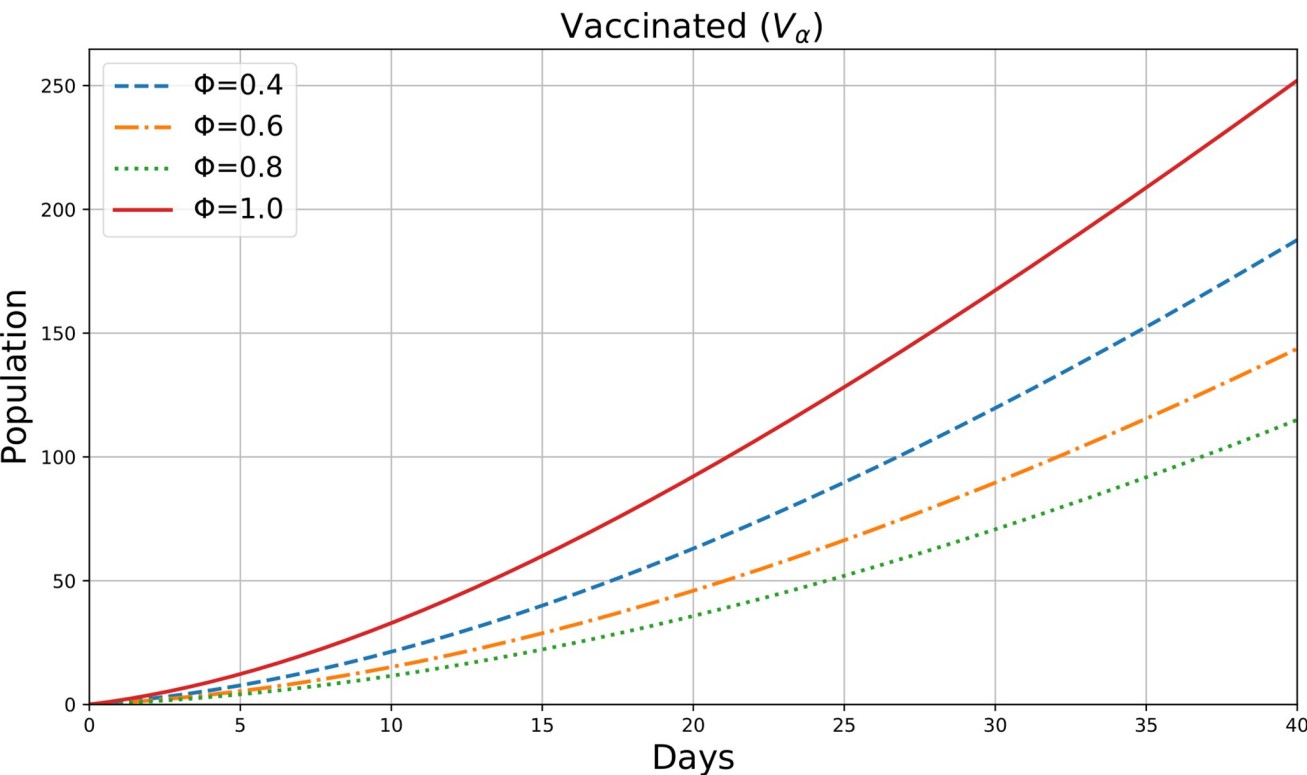

**Fig 9. Vaccinated individuals.**

The variable $E_\alpha(\rho)$ seems to be plotted in this figure for various values of the parameter $\phi$=0.4,0.6,0.8,1, which indicate the number of exposed people in society throughout a period $\rho$ (days). Fig 6 shows the dynamic change in Exposed Individuals in 40 days. The number of exposed individuals initially increases, peaks, and then decreases. Higher $\phi$ values lead to a faster rise and fall in the exposed population. This suggests that with higher $\phi$, exposed individuals either progress to other compartments (like quarantined or infected) more quickly or recover faster.

For various values of the parameter $\phi$=0.4,0.6,0.8,1, this figure appears to be an illustration of the variable $Q_\alpha(\rho)$, which may indicate the number of Quarantined individuals in society during a period $\rho$(days). Fig 7 shows the dynamic change in Quarantined Individuals in 40 days. The quarantined population increases initially and then decreases. Higher $\Phi$ values result in a quicker increase in the quarantined population. This indicates that the fractional derivative parameter accelerates the rate at which exposed individuals are moved to quarantine.

Fig 8 shows the dynamic change in Infected Individuals in 40 days. For various values of the parameter $\phi$=0.4,0.6,0.8,1, this figure appears to be an illustration of the variable $U_\alpha(\rho)$, which may indicate the number of Infected individuals in society during a period $\rho$ (days). After an initial rise, the infected population begins to decrease due to vaccination. Increased $\phi$ values lead the infected population to grow and stabilize more quickly than the quarantined population. This demonstrates how the onset of infection speeds up by fractional derivatives.

For various values of the parameter $\phi$=0.4,0.6,0.8,1, this figure appears to be an illustration of the variable $V_\alpha(\rho)$, which may indicate the number of Vaccinated individuals in society during a period $\rho$(days). Fig 9 shows the dynamic change in Vaccinated Individuals in 40 days.

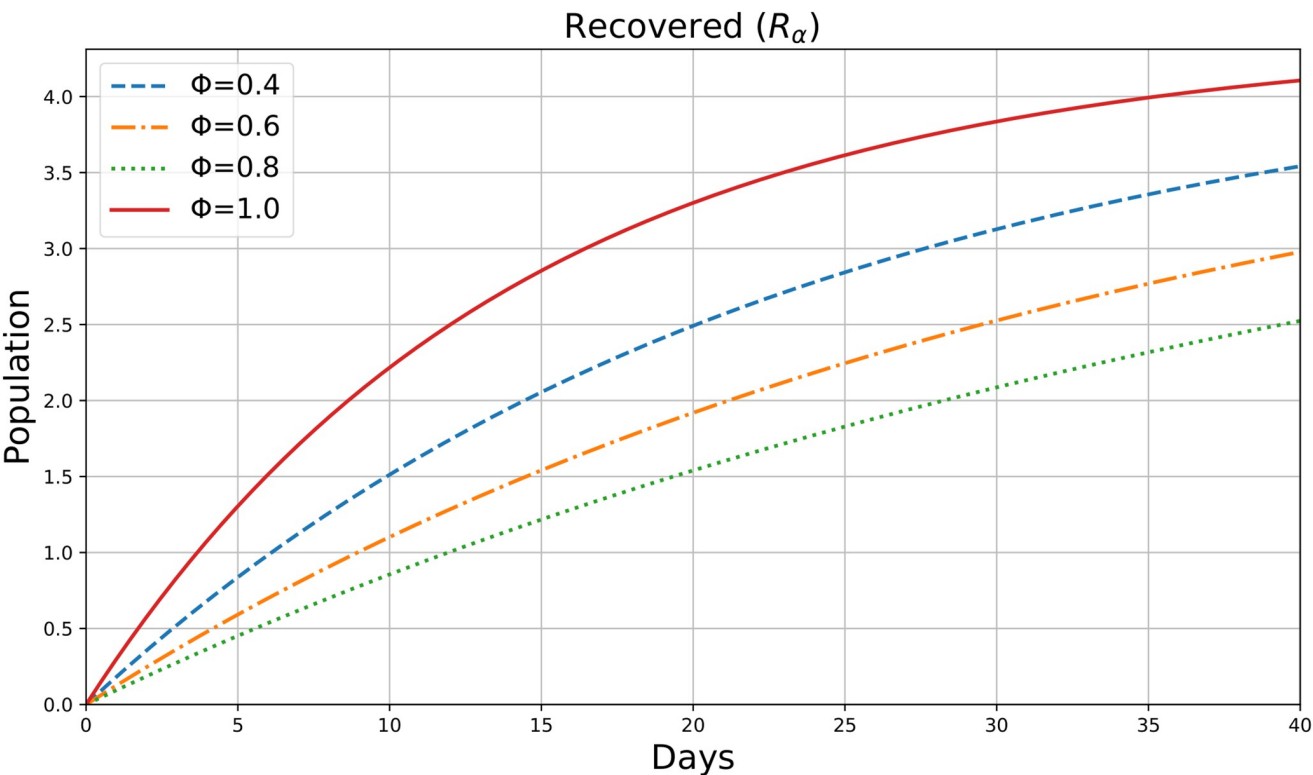

**Fig 10. Recovered individuals.**

These results show that the vaccinated population starts at zero and increases over time so we can say that the chance of infection is reduced. Higher $\phi$ values result in a more rapid increase in the vaccinated population. This suggests that fractional derivatives enhance the vaccination rate or the effectiveness of vaccination over time.

This figure appears to be a plot of the variable $R_\alpha(\rho)$, possibly representing the number of Recovered individuals in society over sometime $\rho$(days), which gives more efficient results for different values of the parameter $\phi=0.4, 0.6, 0.8, 1$. Fig 10 shows the dynamic change in Recovered Individuals in 40 days. These results show that the rate of Recovered people is continuously increasing so we can say that the chance of infection is reduced, and the situation is going to be under control. Higher $\Phi$ values lead to a quicker rise in the recovered population, indicating faster recovery rates when the influence of fractional derivatives is stronger.

### 4.6 Discussion

In our model, conformable fractional derivatives play a pivotal role by incorporating fractional-order dynamics, which blend classical and modern mathematical approaches. The versatility of this derivative lies in its capacity to converge to the solutions of the classical model (1) as the fractional parameter $\phi \rightarrow 1$. This feature highlights the adaptability of the fractional model, ensuring consistency and continuity across different models. Moreover, the high convergence demonstrated by our finite difference technique, as illustrated in Table 3, underscores its efficacy and reliability in computational simulations, especially evidenced by the substantial reduction in error with increasing discretization levels ($N$). Furthermore, Simulations were conducted using the Runge–Kutta fourth-order method which reveal a direct correlation

**Table 3. Comparison of numerical scheme accuracy across different time levels *N* for various values of ϕ.**

| N | $S_\alpha$ | $E_\alpha$ | $Q_\alpha$ | $U_\alpha$ | $V_\alpha$ | $R_\alpha$ |
|---|---|---|---|---|---|---|
| **ϕ = 0.4** | | | | | | |
| 10 | E-02 | E-01 | E-02 | E-01 | E-03 | E-01 |
| 50 | E-03 | E-02 | E-04 | E-02 | E-03 | E-01 |
| 500 | E-05 | E-03 | E-04 | E-02 | E-04 | E-03 |
| 1500 | E-06 | E-04 | E-06 | E-04 | E-05 | E-03 |
| 3000 | E-07 | E-05 | E-07 | E-05 | E-06 | E-04 |
| 5000 | E-09 | E-06 | E-08 | E-06 | E-07 | E-05 |
| 10000 | E-11 | E-08 | E-10 | E-08 | E-09 | E-06 |
| **ϕ = 0.6** | | | | | | |
| 10 | E-03 | E-01 | E-03 | E-01 | E-03 | E-03 |
| 50 | E-04 | E-02 | E-04 | E-02 | E-03 | E-01 |
| 500 | E-05 | E-03 | E-04 | E-02 | E-04 | E-03 |
| 1500 | E-06 | E-04 | E-06 | E-04 | E-05 | E-03 |
| 3000 | E-07 | E-05 | E-07 | E-05 | E-06 | E-04 |
| 5000 | E-08 | E-06 | E-08 | E-06 | E-07 | E-05 |
| 10000 | E-10 | E-08 | E-10 | E-08 | E-09 | E-06 |
| **ϕ = 0.8** | | | | | | |
| 10 | E-03 | E-01 | E-03 | E-01 | E-03 | E-03 |
| 50 | E-04 | E-02 | E-04 | E-02 | E-03 | E-01 |
| 500 | E-05 | E-03 | E-04 | E-02 | E-04 | E-03 |
| 1500 | E-06 | E-04 | E-06 | E-04 | E-05 | E-03 |
| 3000 | E-07 | E-05 | E-07 | E-05 | E-06 | E-04 |
| 5000 | E-08 | E-06 | E-08 | E-06 | E-07 | E-05 |
| 10000 | E-10 | E-08 | E-10 | E-08 | E-09 | E-06 |
| **ϕ = 1** | | | | | | |
| 10 | E-03 | E-01 | E-03 | E-01 | E-03 | E-03 |
| 50 | E-04 | E-02 | E-04 | E-02 | E-03 | E-01 |
| 500 | E-05 | E-03 | E-04 | E-02 | E-04 | E-02 |
| 1500 | E-06 | E-04 | E-05 | E-04 | E-05 | E-02 |
| 3000 | E-07 | E-05 | E-07 | E-05 | E-06 | E-03 |
| 5000 | E-06 | E-06 | E-08 | E-06 | E-07 | E-04 |
| 10000 | E-7 | E-08 | E-9 | E-08 | E-08 | E-05 |

between vaccination rates and infection rates, as depicted in Fig 11. The inverse relationship observed suggests that as vaccination rates increase, infection rates correspondingly decrease. This correlation underscores the pivotal role of vaccination strategies in controlling and mitigating the spread of infectious diseases. It highlights the essential role that vaccination programs perform in preventing the spread of disease and emphasizes their significance as an essential component of public health campaigns.

## 5 Conclusion

In conclusion, our work presents a thorough examination of how vaccine treatment affects COVID-19 dynamics. We have better understood the interaction between vaccination, disease

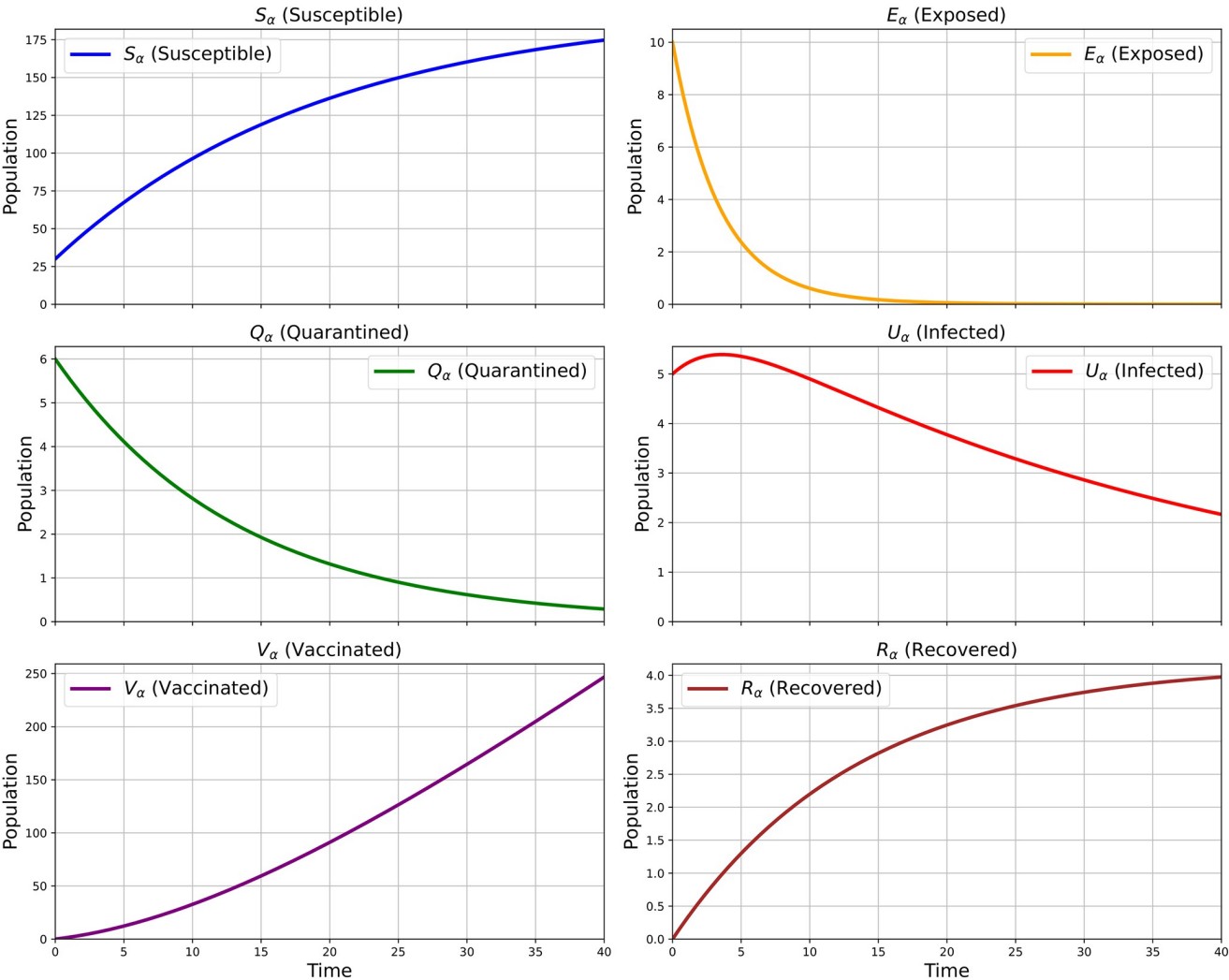

**Fig 11. RecBehavior of COVID-19 model at integer order derivative.**

transmission, and control measures by combining a six-dimensional compartmental model and using conformable derivatives. Our findings emphasize the significance of including vaccination and quarantined strategies in mathematical models since they considerably impact the disease's overall dynamics and the fundamental reproduction number ($R_0$). The stability studies of the above equilibrium points have been examined in the following, and it has been demonstrated that the DFE is asymptotically stable when $R_0 < 1$ and unstable when $R_0 > 1$. We designed a finite difference approach for the conformable fractional derivative using the Taylor series achieving a highly convergent solution for the system of equations. We have calculated the efficiency of vaccination in preventing the spread of COVID-19 using careful mathematical simulations and sensitivity analysis. The boundedness, positiveness, and positiveness of the solutions have all been established. We have demonstrated the existence and uniqueness of the solutions using the Lipschitz condition. Future work may explore using another fractional derivative on a modified mathematical model of experimental data.

## Author Contributions

**Conceptualization:** Tamour Zubair.

**Formal analysis:** Tamour Zubair, Sehrish Ramzan.

**Funding acquisition:** Muhammad Bilal Riaz, Taseer Muhammad.

**Investigation:** Tamour Zubair, Sehrish Ramzan.

**Project administration:** Muhammad Imran Asjad.

**Resources:** Syeda Alishwa Zanib, Sehrish Ramzan.

**Software:** Syeda Alishwa Zanib, Tamour Zubair.

**Supervision:** Tamour Zubair.

**Validation:** Syeda Alishwa Zanib, Muhammad Imran Asjad, Taseer Muhammad.

**Visualization:** Muhammad Bilal Riaz.

**Writing – original draft:** Syeda Alishwa Zanib, Sehrish Ramzan.

**Writing – review & editing:** Muhammad Imran Asjad, Taseer Muhammad.

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
