## [Decision Letter · Decision Letter 0]

13 Nov 2023

PONE-D-23-25335A Conformable Fractional Finite Difference Method for

Modified Mathematical Modeling of SAR-CoV-2

(COVID-19) DiseasePLOS ONE

Dear Dr. Zubair,

Thank you for submitting your manuscript to PLOS ONE. After careful consideration, we feel that it has merit but does not fully meet PLOS ONE’s publication criteria as it currently stands. Therefore, we invite you to submit a revised version of the manuscript that addresses the points raised during the review process.

We look forward to receiving your revised manuscript.

Kind regards,

Renier Mendoza

Academic Editor

PLOS ONE

“NO”

5. Please update your submission to use the PLOS LaTeX template. The template and more information on our requirements for LaTeX submissions can be found at http://journals.plos.org/plosone/s/latex.

Reviewers' comments:

Reviewer's Responses to Questions

**Comments to the Author**

1. Is the manuscript technically sound, and do the data support the conclusions?

Reviewer #1: Yes

Reviewer #2: Partly

Reviewer #3: Yes

2. Has the statistical analysis been performed appropriately and rigorously? 

Reviewer #1: Yes

Reviewer #2: N/A

Reviewer #3: Yes

3. Have the authors made all data underlying the findings in their manuscript fully available?

Reviewer #1: Yes

Reviewer #2: No

Reviewer #3: Yes

4. Is the manuscript presented in an intelligible fashion and written in standard English?

Reviewer #1: Yes

Reviewer #2: No

Reviewer #3: Yes

5. Review Comments to the Author

Reviewer #1: In the present manuscript investigate modified and comprehensive mathematical model that captures the complex relationships between various population compartments, including susceptible, infected, exposed, recovered, vaccinated, and quarantined people. Using conformable derivatives, we provide a system of equations that precisely captures the complex interconnections inside the COVID-19 transmission.

This field also has a wide applications. Clearly organized in analytical as well as numerical part. The work of the paper is good. However, there are some comments to improve the quality of the paper which are given as follows:

[1.] Model formation is unclear, explain it well.

[2.] Explain novelty or key contributions of your work in the end of introduction part.

[3.] There are lot of grammatical error, some typos and punctuation problems are exist that should be checked and corrected throughout the paper.

[4.] What about the solution of system?

[5.] What is the main objective of the research presented in the abstract, and why is it important in the context of the COVID-19 pandemic?

[6.] How does the mathematical model described in the abstract account for the different population compartments, and what role do conformable derivatives play in this model?

[7.] What methods were used for performing calculations in this research, and why was MAPLE-2019 chosen for these calculations?

[8.] How you fix the initial values of the model?

[9.] In Introduction part, Start with mathematical modeling and add some important works related with recent development in mathematical modeling such as Biomedical Signal Processing and

Control, 84, 2023, 104714; Results in Physics, 2023, 106489; Nonlinear Dynamics, 111 4879-4914, 2023; Symmetry, 15(4), 845, 2023; Optimal Control, Applications and Methods, (43)(3), 842-866, 2022; Nonlinear Dynamics, 109 3169{3187, 2022.

Reviewer #2: The manuscript presented a mathematical model using conformable fractional derivatives for COVID-19 with quarantine and vaccination. The authors claim that by using conformable derivatives, the resulting system of equations “precisely captures” the complex interconnections in the COVID-19 transmission. While I applaud the authors’ efforts in this regard, the paper does not provide sufficient analysis and/or numerical results to support this claim. I suggest that the authors revisit their work and hire a copyeditor.

Some guide questions/suggestions that may help improve their work:

- Why use conformable derivatives over other fractional derivatives? What are the advantages?

- Are there drawbacks in using conformable derivatives in comparison to other fractional derivatives and ordinary derivatives?

- In what sense does the proposed model based on systems of conformal differential equations “precisely captures” the complex interconnections in the COVID-19 transmission compared to models based on ODE systems or other FDE systems?

- What is the main contribution of your work? How does it compare to similar works, e.g. the fractal-fractional model for COVID-19, proposed by Malik et al. (https://doi.org/10.1016/j.aej.2022.02.024), where they also incorporated quarantine and vaccination?

Moreover, the literature review in the introduction is quite lengthy, but it mainly focuses on COVID-19 and models based on ODE systems. Related literature on fractional derivatives and mathematical models based on systems of fractional differential equations should be reported in the manuscript. The research objectives should be stated in the introduction. Lastly, it is should be noted that this year 2023, the World Health Organization has declared an end to COVID-19 as a global health emergency, and with the overwhelmingly large number of literature on mathematical models for COVID-19, it is crucial that the authors highlight the main contribution of their work in this vast field.

Unfortunately, in its current state, I believe the material is too underdeveloped. For these reasons, I cannot recommend accepting this paper for publication.

Reviewer #3: Authors have constructed novel results. I recommend but need major concerns like:

1. Check the stability of the method.

2. Why existence uniqueness is considered. For such existence refer original work like:On nonlinear conformable fractional order dynamical system via differential transform method." CMES-Computer Modeling in Engineering & Sciences 136.2 (2023): 1457-1472.Existence and stability theory of pantograph conformable fractional differential problem." Thermal Science 27.Spec. issue 1 (2023): 237-244.Mathematical analysis of fractional order alcoholism model." Alexandria Engineering Journal 78 (2023): 281-291.

3. Discuss the convergence of the numerical method.

4. Why this method is powerful than other refer some work like: The Volterra-Lyapunov matrix theory and nonstandard finite difference scheme to study a dynamical system." Results in Physics 52 (2023): 106890.Study of transmission dynamics of novel COVID-19 by using mathematical model." Advances in Difference Equations 2020 (2020): 1-13.To study the transmission dynamic of SARS-CoV-2 using nonlinear saturated incidence rate." Physica A: Statistical Mechanics and its Applications 604 (2022): 127915.

4.Discuss graphs biologically.

5.Update the conclusion with future work.

6. PLOS authors have the option to publish the peer review history of their article (what does this mean?). If published, this will include your full peer review and any attached files.

Reviewer #1: No

Reviewer #2: No

Reviewer #3: No

---

## [Author Response · Author response to Decision Letter 0]

12 Feb 2024

Response to reviewer’s comments

Manuscript ID: PONE-D-23-25335

Journal Name: PLOS One

Dear Associate Editor,

Thank you for your useful comments on our manuscript. We have considered your editorial or reviewer's comments 

and made the following changes to our paper entitled "A Conformable Fractional Finite Difference Method for 

Modified Mathematical Modeling of SAR-CoV-2 (COVID-19) Disease." We have modified the manuscript 

accordingly, and detailed corrections are listed below point by point. All the changes are highlighted in red, blue, 

and green color for Referees 1, 2 and, 3 respectively, in the paper.

Comments from Reviewers and Answers

Reviewer: I

In the present manuscript investigate modified and comprehensive mathematical model that captures the complex 

relationships between various population compartments, including, susceptible, exposed, recovered, vaccinated, and 

quarantined people. Using Conformable derivatives, we provide a system of equations that precisely captures the 

complex interconnections inside the COVID-19 transmission. This field also have a wide application. Clearly 

organized in analytical as well as numerical part. The work of the paper is good. However, there are some comments 

to improve the quality of paper which are given as follows:

1. Model formation is unclear, explain it well.

Response. The detail model formulation has been added

2. Explain the novelty or key contributions of your work at the end of the introduction part.

Response. Query has been fixed

3. There are lot of grammatical errors, some typos and punctuation problems are exist that should be checked and 

corrected throughout the paper.

Response. All grammar errors have been removed accordingly

4. What about the solution of the system?

Response. We calculate the solution of the system using the finite difference method.

5. What is the main objective of the research presented in the abstract, and why is the important in the context of the 

COVID-19 pandemic?

Response. The query has been fixed.

6. How does the mathematical model described in the abstract account for the different population compartments, 

and what role do conformable derivatives play in the model?

Response. The mathematical model described in the abstract incorporates different population compartments by 

utilizing a six-dimensional compartmental model. Conformable derivatives play a crucial role in the model by 

introducing fractional-order dynamics.

7. What method used for performing calculations in this research, and why was MAPLE-19 chosen for these 

calculations?

Response: FDM is used for calculation. MAPLE-19 was chosen as the tool for these calculations because it's like a 

super-smart calculator. It's good at working with symbols and equations, which is important when dealing with 

complex math.

8. How you fix the initial values of model?

Response: Initial conditions have been set using literature.

9. In introduction part, start with mathematical modelling and add some important works related with recent 

development in mathematical modelling such as biomedical signal processing and control, 84, 2023, 104714;

Results in Physics, 2023, 106489; Nonlinear Dynamics,111, 4879-4914, 2023; Symmetry, 15(4), 854, 2023; optimal 

control, Applications and Methods, (43)(3), 842-866, 2022; Nonlinear Dynamics, 109 3169(3187),2022.

Response: The introduction has been modified using the suggested reference.

Reviewer II

The manuscript a mathematical model using conformable fractional derivatives for Covid-19 with quarantine and 

vaccinations, The authors claim that by using conformable derivatives, the resulting system of equations precisely 

captures the complex interconnections in COVID-19 transmission. While I applaud the author’s efforts in this regard 

paper does not provide sufficient analysis and/or numerical results to support this claim. I suggest that the authors 

revisit their work and hire a copyeditor. Some guide questions/suggestions that may help to improve their work:

1. why use conformable derivatives over other fractional derivatives? What are the advantages?

Response: Conformable derivatives play a crucial role in the model by introducing fractional-order dynamics.

The beauty of the conformable fractional order derivative is that the solution of the fractional model (9)-(14), tends 

to the solution of the classical model (1)-(6), as the value of 𝜙 tends to 1.

2. Are there drawbacks in using conformable derivatives in comparison to other fractional derivatives and ordinary 

derivatives? 

Response: Conformable derivatives are simpler but may not work in all situations other than fractional and ordinary 

derivatives.

3. In what sense does the proposed model based on systems of conformal differential equations precisely capture the 

complex interconnections in the COVID-19, transmission compared to models based on ODE systems or other FDM 

systems?

Response: Applying conformable fractional derivatives to a COVID-19 mathematical model enhances sensitivity to 

variations, and incorporates memory effects resulting in a more accurate representation of the virus's complex 

dynamics compared to other models based on ODE systems or FDM systems.

4. What is mean contribution of your work? How does it compare to similar works. e.g the fractal-fractional model 

for COVID-19 proposed by Malik et al., (https://doi.org/10.1016/j.aej.2022.02.024), where they also incorporated 

quarantine and vaccinations?

Response: We designed the FDM approach of the conformable fractional derivative using the Taylor series and 

used it on a modified mathematical model of COVID-19 which is the main contribution of the work.

5. Moreover the literature review in the introduction is quite lengthy, but it mainly focuses on COVID-19 and model 

based on ODE systems. Related literature on fractional derivatives and mathematical models based on systems of 

fractional differential equations should be represented in the manuscript. The research objectives should be stated in 

the introduction. 

Response: The literature review has been updated as suggested.

6. Lastly, it is should be noted that this year 2023, the world health organization has declared an end to COVID-19 

as a global health emergency and with overwhelmingly large number of literature on mathematical models for 

COVID-19, it is crucial that the authors highlights the main contribution of their work in the vast field.

Unfortunately, in the current state, I believe that the material is too underdeveloped. For this reason, I cannot 

recommend accepting this paper for publication.

Response: As mentioned already the main focus of the work is the FDM approach of the conformable fractional 

derivative using the Taylor series and used on a modified mathematical model of COVID-19

Reviewer III

Authors have constructed novel results. I recommend but need major concerns like:

1. Check the stability of the method.

Response: We check the stability of the model at a disease-free equilibrium point. In the numerical section, we also 

check the stability analysis of the numerical scheme

2. Why existences uniqueness is considered. For such existence refer original work like : on nonlinear conformable 

fractional order dynamical system via differential transform method. CMES.Computer Modeling in Engerning & 

science 136.2 (2023): 1457-1472. Existence and stability theory of pantograph conformable fractional differential 

problem. Thermal Science 27.spec. issue 1 (2023): 237-244. Mathematical analysis of fractional order alcoholism 

model. Alexandria Engerning journal 78 (2023): 281-291.

Response: Existence and uniqueness considerations in mathematical models ensure that solutions are well-defined 

and meaningful. For the method, suitable reference has been cited.

3. Discus the convergence of the numerical method.

Response: Convergence of numerical scheme has been discussed.

4. Why this method is powerful than other refer work like. The Volterra-Lyapunov matrix theory and non-understand finite difference scheme to study a dynamical system. Results in Physics 52 (2023): 106890. Study a 

transmission dynamical of novel COVID-19 by using a mathematical model. Advanced in differential equations 

2020(2020): 1-13. To study the transmission dynamics of SARS-COV-2 using a nonlinear saturated incidence rate. 

Physics A: Statistical Mechanics and its Applications 604 (2022): 127915:

Response: The conformable fractional derivative method is better than other methods because it can understand and 

represent more complex behaviors in systems, making it a more flexible and accurate tool for studying dynamic 

systems compared to the Volterra-Lyapunov matrix theory.

5. Discuss graphs biologically.

Response: Biological Behavior of graphs have been discussed.

6. Update the conclusion with future fork

Response: Future directions have been added in conclusion

Response to Associate Editor:

The authors would like to thank the referees for suggesting specific changes in the original manuscript, for their 

valuable comments which improved the paper and for their high interest in this work. The manuscript has been 

resubmitted to your journal. We look forward to your positive response.

With Regards,

Tamour Zubair

---

## [Decision Letter · Decision Letter 1]

20 Mar 2024

PONE-D-23-25335R1A Conformable Fractional Finite Difference Method for Modified Mathematical Modeling of SAR-CoV-2 (COVID-19) DiseasePLOS ONE

Dear Dr. Zubair,

Thank you for submitting your manuscript to PLOS ONE. After careful consideration, we feel that it has merit but does not fully meet PLOS ONE’s publication criteria as it currently stands. Therefore, we invite you to submit a revised version of the manuscript that addresses the points raised during the review process. **Please refer to the comments and suggestions below as you revise the manuscript.**

We look forward to receiving your revised manuscript.

Kind regards,

Renier Mendoza

Academic Editor

PLOS ONE

Journal Requirements:

Additional Editor Comments:

During the initial revision of the manuscript, THREE additional authors were included. In your response letter, please specify the sections of the manuscript to which each author contributed and explain the significance of their contributions. Additionally, please explain why the original authors are unable to revise the manuscript independently without the assistance of the new authors. Failure to provide justification for the inclusion of these authors may lead to rejection.

Reviewers' comments:

Reviewer's Responses to Questions

**Comments to the Author**

1. If the authors have adequately addressed your comments raised in a previous round of review and you feel that this manuscript is now acceptable for publication, you may indicate that here to bypass the “Comments to the Author” section, enter your conflict of interest statement in the “Confidential to Editor” section, and submit your "Accept" recommendation.

Reviewer #1: (No Response)

Reviewer #2: All comments have been addressed

Reviewer #3: (No Response)

2. Is the manuscript technically sound, and do the data support the conclusions?

Reviewer #1: Yes

Reviewer #2: Partly

Reviewer #3: Yes

3. Has the statistical analysis been performed appropriately and rigorously? 

Reviewer #1: Yes

Reviewer #2: N/A

Reviewer #3: Yes

4. Have the authors made all data underlying the findings in their manuscript fully available?

Reviewer #1: Yes

Reviewer #2: Yes

Reviewer #3: Yes

5. Is the manuscript presented in an intelligible fashion and written in standard English?

Reviewer #1: Yes

Reviewer #2: No

Reviewer #3: No

6. Review Comments to the Author

Reviewer #1: The authors' revisions have been duly noted. Consequently, I hereby accept the manuscript. Their diligent updates have enhanced its quality, addressing concerns and refining content. Their efforts reflect a commitment to scholarly excellence, ensuring the work meets publication standards. Acknowledging their dedication, I endorse the manuscript for inclusion. This decision underscores its significance and contribution to the field. I anticipate its impact on readers and its role in advancing knowledge. With confidence in its merit, I affirm my acceptance of the manuscript, appreciating the authors' contributions and the collective endeavor towards academic advancement.

Reviewer #2: The authors were able to address some comments in the first revision. In my opinion, the paper can be accepted after revision taking into consideration the following points.

1. An in-depth discussion in the paper is necessary to establish why the proposed model based on conformable fractional derivative is better than classical models based on ODE systems. These needs to be highlighted as well in the numerical results.

2. The paper claims that the use of a finite difference approach over classical numerical methods gives "high convergence", but the paper fails to satisfatorily establish this, both theoretically and numerically.

3. There are still grammatical, typographical, punctuation, and typesetting issues. I urge the authors to hire a copyeditor to help address these issues.

Reviewer #3: Authors have not addressed all my comments.

Graphical illustrations need explanations. Typos exist there.

Still the literature need to update about the NSFD method. It is strongly recommend to put relevant references about the methodology been used for COVID-19. Authors have ignored these . They must cite the research work relevant to COVID-19 recently available like: Study of a mathematical model of COVID-19 outbreak using some advanced analysis." Waves in Random and Complex Media (2022): 1-18.Study of integer and fractional order COVID-19 mathematical model." Fractals (2023): 2340046.Fractional order modeling of predicting covid-19 with isolation and vaccination strategies in morocco." CMES-Comput. Model. Eng. Sci 136 (2023): 1931-1950.On a SEIR-type model of COVID-19 using piecewise and stochastic differential operators undertaking management strategies." (2023).

For NSFD cite these work: Results in Physics 52 (2023): 106890, Results in Physics 24 (2021): 104069.

Update the literature about applications of mathy. models like: Statistical and computational analysis for corruption and poverty model using Caputo-type fractional differential equations." Heliyon (2024)., A Fractal-Fractional Order Model to Study Multiple Sclerosis: A Chronic Disease." Fractals (2024): 2440010. South African Journal of Chemical Engineering 48 (2024): 63-70.

7. PLOS authors have the option to publish the peer review history of their article (what does this mean?). If published, this will include your full peer review and any attached files.

Reviewer #1: No

Reviewer #2: No

Reviewer #3: No

---

## [Author Response · Author response to Decision Letter 1]

2 May 2024

Response to reviewer’s comments

Manuscript ID: PONE-D-23-25335

Journal Name: PLOS One

Dear Associate Editor,

Thank you for your useful comments on our manuscript. We have considered your editorial or reviewer's comments and made the following changes to our paper entitled " A Conformable Fractional Finite Difference Method for Modified Mathematical Modeling of SAR-CoV-2 (COVID-19) Disease". We have modified the manuscript accordingly, and detailed corrections are listed below point by point. All the changes are highlighted in green, blue, and red color for Referee 1, Referee 2, and Referee 3, respectively in the paper. 

Comments from Reviewers and Answers

Reviewer: I

The authors' revisions have been duly noted. Consequently, I hereby accept the manuscript. Their diligent updates have enhanced its quality, addressing concerns and refining content. Their efforts reflect a commitment to scholarly excellence, ensuring the work meets publication standards. Acknowledging their dedication, I endorse the manuscript for inclusion. This decision underscores its significance and contribution to the field. I anticipate its impact on readers and its role in advancing knowledge. With confidence in its merit, I affirm my acceptance of the manuscript, appreciating the authors' contributions and the collective endeavor toward academic advancement.

Response: We're grateful to you for your positive feedback and acceptance of our revised manuscript.

Reviewer: II

 The authors were able to address some comments in the first revision. In my opinion, the paper can be accepted after revision taking into consideration the following points.

1. An in-depth discussion in the paper is necessary to establish why the proposed model based on conformable fractional derivative is better than classical models based on ODE systems. These need to be highlighted as well in the numerical results.

Response: We appreciate the reviewer's suggestion and agree that an in-depth discussion is crucial. In our revised manuscript, we will elaborate conformable fractional derivative is better than classical models based on ODE systems. We will also enhance our numerical results section to include a comparative analysis demonstrating how our fractional derivative model provides improved accuracy and insight into system behaviours as the fractional parameter ϕ approaches 1, thereby bridging the understanding between classical and fractional models in the discussion section.

2. The paper claims that the use of a finite difference approach over classical numerical methods gives "high convergence", but the paper fails to satisfactorily establish this, both theoretically and numerically.

Response: Table 3 explains that the finite difference approach gives high convergence results.

3. There are still grammatical, typographical, punctuation, and typesetting issues. I urge the authors to hire a copyeditor to help address these issues.

Response: All grammatical, typographical, punctuation, and typesetting issues have been removed.

Reviewer: III

The authors have not addressed all my comments.

1. Graphical illustrations need explanations. Typos exist there.

Response:

Thank you for your kind suggestion. Graphical explanations have been added and remove all typos.

2. Still, the literature needs to be updated about the NSFD method. It is strongly recommended to put relevant references about the methodology used for COVID-19. Authors have ignored these. They must cite the research work relevant to COVID-19 recently available like: Study of a mathematical model of COVID-19 outbreak using some advanced analysis." Waves in Random and Complex Media (2022): 1-18. Study of integer and fractional order COVID-19 mathematical model." Fractals (2023): 2340046.Fractional order modeling of predicting covid-19 with isolation and vaccination strategies in Morocco." CMES-Comput. Model. Eng. Sci 136 (2023): 1931-1950.On an SEIR-type model of COVID-19 using piecewise and stochastic differential operators undertaking management strategies." (2023).

For NSFD cite these work: Results in Physics 52 (2023): 106890, Results in Physics 24 (2021): 104069.

Update the literature about applications of mathy. models like: Statistical and computational analysis for corruption and poverty model using Caputo-type fractional differential equations." Heliyon (2024)., A Fractal-Fractional Order Model to Study Multiple Sclerosis: A Chronic Disease." Fractals (2024): 2440010. South African Journal of Chemical Engineering 48 (2024): 63-70.

Response: All suitable references have been cited according to instructions.

Response to Associate Editor:

The authors would like to thank the referees for suggesting specific changes in the original manuscript, for their valuable comments which improved the paper and for their high interest in this work. The manuscript has been resubmitted to your journal. We look forward to your positive response.

With Regards,

Tamour Zubair

---

## [Editor Report · Decision Letter 2]

7 May 2024

PONE-D-23-25335R2A Conformable Fractional Finite Difference Method for

Modified Mathematical Modeling of SAR-CoV-2

(COVID-19) Disease

PLOS ONE

Dear Dr. Zubair,

Thank you for submitting your manuscript to PLOS ONE. After careful consideration, we feel that it has merit but does not fully meet PLOS ONE’s publication criteria as it currently stands. Therefore, we invite you to submit a revised version of the manuscript that addresses the points raised during the review process.

Specifically, the authors did not address a very important point. During the initial revision of the manuscript, THREE additional authors were added. I am sending the manuscript back to the authors before sending it again for review. Please revise your response letter and specify the sections of the manuscript to which each author contributed and explain the significance of their contributions. Additionally, please explain why the original authors are unable to revise the manuscript independently without the assistance of the new authors. **Failure to provide justification for the inclusion of these authors during the revision may lead to rejection. **PLOS is a member of the Committee on Publication Ethics (COPE). PLOS ONE abides by its Code of Conduct and aims to adhere to its Best Practice Guidelines. **Authors are expected to comply with best practices in publication ethics, specifically concerning authorship**, dual publication, plagiarism, figure manipulation, and competing interests.

We look forward to receiving your revised manuscript.

Kind regards,

Renier Mendoza

Academic Editor

PLOS ONE
---

## [Author Response · Author response to Decision Letter 2]

27 May 2024

Response to reviewer’s comments

Manuscript ID: PONE-D-23-25335

Journal Name: PLOS One

Dear Associate Editor,

Thank you for your useful comments on our manuscript. We have considered your editorial or 

reviewer's comments and made the following changes to our paper entitled " A Conformable 

Fractional Finite Difference Method for Modified Mathematical Modeling of SAR-CoV-2 

(COVID-19) Disease". We have modified the manuscript accordingly, and detailed corrections are 

listed below point by point. All the changes are highlighted in green, blue, and red color for Referee

1, Referee 2, and Referee 3, respectively in the paper.

Comments from Reviewers and Answers

Reviewer: I

The authors' revisions have been duly noted. Consequently, I hereby accept the manuscript. Their 

diligent updates have enhanced its quality, addressing concerns and refining content. Their efforts 

reflect a commitment to scholarly excellence, ensuring the work meets publication standards. 

Acknowledging their dedication, I endorse the manuscript for inclusion. This decision underscores 

its significance and contribution to the field. I anticipate its impact on readers and its role in 

advancing knowledge. With confidence in its merit, I affirm my acceptance of the manuscript, 

appreciating the authors' contributions and the collective endeavor toward academic advancement.

Response: We're grateful to you for your positive feedback and acceptance of our revised 

manuscript.

Reviewer: II

The authors were able to address some comments in the first revision. In my opinion, the paper 

can be accepted after revision taking into consideration the following points.

1. An in-depth discussion in the paper is necessary to establish why the proposed model based on 

conformable fractional derivative is better than classical models based on ODE systems. These 

need to be highlighted as well in the numerical results.

Response: We appreciate the reviewer's suggestion and agree that an in-depth discussion is 

crucial. In our revised manuscript, we will elaborate conformable fractional derivative is better 

than classical models based on ODE systems. We will also enhance our numerical results section

4 to include a comparative analysis demonstrating how our fractional derivative model provides 

improved accuracy and insight into system behaviours as the fractional parameter 𝜙 approaches 1, 

thereby bridging the understanding between classical and fractional models in the discussion 

section.

2. The paper claims that the use of a finite difference approach over classical numerical methods 

gives "high convergence", but the paper fails to satisfactorily establish this, both theoretically and 

numerically.

Response: Table 3 explains that the finite difference approach gives high convergence results.

3. There are still grammatical, typographical, punctuation, and typesetting issues. I urge the authors 

to hire a copyeditor to help address these issues.

Response: All grammatical, typographical, punctuation, and typesetting issues have been 

removed.

Reviewer: III

The authors have not addressed all my comments.

1. Graphical illustrations need explanations. Typos exist there.

Response:

Thank you for your kind suggestion. Graphical explanations have been added and remove all 

typos.

2. Still, the literature needs to be updated about the NSFD method. It is strongly recommended to 

put relevant references about the methodology used for COVID-19. Authors have ignored these. 

They must cite the research work relevant to COVID-19 recently available like: Study of a 

mathematical model of COVID-19 outbreak using some advanced analysis." Waves in Random 

and Complex Media (2022): 1-18. Study of integer and fractional order COVID-19 mathematical 

model." Fractals (2023): 2340046.Fractional order modeling of predicting covid-19 with isolation 

and vaccination strategies in Morocco." CMES-Comput. Model. Eng. Sci 136 (2023): 1931-

1950.On an SEIR-type model of COVID-19 using piecewise and stochastic differential operators 

undertaking management strategies." (2023).

For NSFD cite these work: Results in Physics 52 (2023): 106890, Results in Physics 24 (2021): 

104069.

Update the literature about applications of mathy. models like: Statistical and computational 

analysis for corruption and poverty model using Caputo-type fractional differential equations." 

Heliyon (2024)., A Fractal-Fractional Order Model to Study Multiple Sclerosis: A Chronic 

Disease." Fractals (2024): 2440010. South African Journal of Chemical Engineering 48 (2024): 

63-70.

Response: Thank you for your suggestions. All suitable references have been cited according to 

instructions.

Response to Associate Editor:

The authors would like to thank the referees for suggesting specific changes in the original

manuscript, for their valuable comments which improved the paper and for their high interest in 

this work. The manuscript has been resubmitted to your journal. We look forward to your positive 

response.

With Regards,

Tamour Zubair

Response Letter

Respected Editor,

We appreciate your feedback and the opportunity to revise our manuscript. We have thoroughly 

considered your comments regarding the inclusion of additional authors and would like to provide 

the necessary clarifications and justifications:

Contributions of the Original Authors:

• Syeda Alishwa Zanib: Originated the manuscript's main idea. Responsibilities included 

writing the original draft, review and editing, methodology development, and 

conceptualization.

• Tamour Zubair: Contributed to review and editing, methodology enhancement, and 

software development.

• Sehrish Ramzan: Focused on methodology development, software support, and 

manuscript review.

Contributions of the Additional Authors:

• Muhammad Bilal Riaz: Contributed to the graphical Analysis. Muhammad Bilal Riaz 

provided significant expertise in advanced statistical and numerical methods, which were 

crucial for the thorough analysis presented in Section 3.3 and 4 of the manuscript. Their 

contribution has enhanced the reliability of the results.

• Muhammad Imran Asjad: Played a pivotal role in the statistical and numerical analysis 

of design fractional method. Muhammad Imran Asjad 's expertise in fieldwork and 

fractional methodologies was indispensable for the accuracy and comprehensiveness of the 

data presented in Section 4.5. Their involvement ensured that the design method adhered 

to the highest standards of scientific rigor.

• Taseer Muhammad: Assisted in the revision of the discussion and conclusion sections. 

Taseer Muhammad brought a fresh perspective and contributed to a more in-depth 

interpretation of the results, particularly in Sections 4.1 and 5. Their insights helped to 

contextualize the findings within the broader scope of existing literature and proofread.

The decision to include these additional authors was based on their substantial contributions to key 

sections of the manuscript, which were beyond the expertise of the original authors. The revisions 

required a multidisciplinary approach to ensure the manuscript's quality and integrity, and the 

added expertise of Muhammad Bilal Riaz, Muhammad Imran Asjad, and Taseer Muhammad was 

essential to meet these high standards. 

The original authors recognized that the complexity of the revisions necessitated specialized 

knowledge and skills that the new authors possess. Their collaboration has been instrumental in 

addressing the reviewers' comments comprehensively and improving the overall quality of the 

manuscript. 

We assure you that the inclusion of these authors is in full compliance with the best practices in 

publication ethics, as they have each made significant and direct contributions to the work. We 

believe that their involvement is fully justified and that their expertise has greatly enhanced the 

manuscript. Thank you for your consideration. 

Sincerely,

Tamour Zubair

---

## [Editor Report · Decision Letter 3]

4 Jun 2024

PONE-D-23-25335R3A Conformable Fractional Finite Difference Method for

Modified Mathematical Modeling of SAR-CoV-2

(COVID-19) DiseasePLOS ONE

Dear Dr. Zubair,

Thank you for submitting your manuscript to PLOS ONE. After careful consideration, we feel that it has merit but does not fully meet PLOS ONE’s publication criteria as it currently stands. Therefore, we invite you to submit a revised version of the manuscript that addresses the points raised during the review process.

Kindly refer to the comments below.

We look forward to receiving your revised manuscript.

Kind regards,

Renier Mendoza

Academic Editor

PLOS ONE

Additional Editor Comments:

In your response, you mentioned that Muhammad Bilal Riaz provided significant expertise in advanced statistical and numerical methods, which were crucial for the thorough analysis presented in Section 3.3. However, Section 3.3 details the computation of the DFEP, and the computed equilibrium did not change after Riaz joined as an author. You also mentioned that Taseer Muhammad contributed to a more in-depth interpretation of the results in Section 4.1. However, upon closer inspection, theorem 4.1 and its proof did not change after Muhammad joined as an author. The justification for the addition of these authors is not satisfactory. Based on your response, the new authors' contributions are not sufficient for their inclusion in the manuscript. Please revise your response or reconsider adding the new authors.

---

## [Author Response · Author response to Decision Letter 3]

15 Jun 2024

There was no query from the referees. That is why no response letter to the referees are attached this time.

 Response to the editor letter is attached to the file to clarify the editor concerns.

---

## [Decision Letter · Decision Letter 4]

10 Jul 2024

A Conformable Fractional Finite Difference Method for

Modified Mathematical Modeling of SAR-CoV-2

(COVID-19) Disease

PONE-D-23-25335R4

Dear Dr. Zubair,

We’re pleased to inform you that your manuscript has been judged scientifically suitable for publication and will be formally accepted for publication once it meets all outstanding technical requirements.

Kind regards,

Renier Mendoza

Academic Editor

PLOS ONE

Additional Editor Comments (optional):

Reviewers' comments:

Reviewer's Responses to Questions

**Comments to the Author**

1. If the authors have adequately addressed your comments raised in a previous round of review and you feel that this manuscript is now acceptable for publication, you may indicate that here to bypass the “Comments to the Author” section, enter your conflict of interest statement in the “Confidential to Editor” section, and submit your "Accept" recommendation.

Reviewer #3: All comments have been addressed

2. Is the manuscript technically sound, and do the data support the conclusions?

Reviewer #3: Yes

3. Has the statistical analysis been performed appropriately and rigorously? 

Reviewer #3: Yes

4. Have the authors made all data underlying the findings in their manuscript fully available?

Reviewer #3: Yes

5. Is the manuscript presented in an intelligible fashion and written in standard English?

Reviewer #3: Yes

6. Review Comments to the Author

Reviewer #3: Acceptable in current form.Acceptable in current form.Acceptable in current form.Acceptable in current form.Acceptable in current form.Acceptable in current form.Acceptable in current form.Acceptable in current form.Acceptable in current form.

7. PLOS authors have the option to publish the peer review history of their article (what does this mean?). If published, this will include your full peer review and any attached files.

Reviewer #3: No

---

## [Editor Report · Acceptance letter]

27 Aug 2024

PONE-D-23-25335R4 

PLOS ONE

Dear Dr. Zubair, 

I'm pleased to inform you that your manuscript has been deemed suitable for publication in PLOS ONE. Congratulations! Your manuscript is now being handed over to our production team.

Kind regards, 

on behalf of

Dr. Renier Mendoza 

Academic Editor

PLOS ONE